# RobuQ: Pushing DiTs to W1.58A2 via Robust Activation Quantization

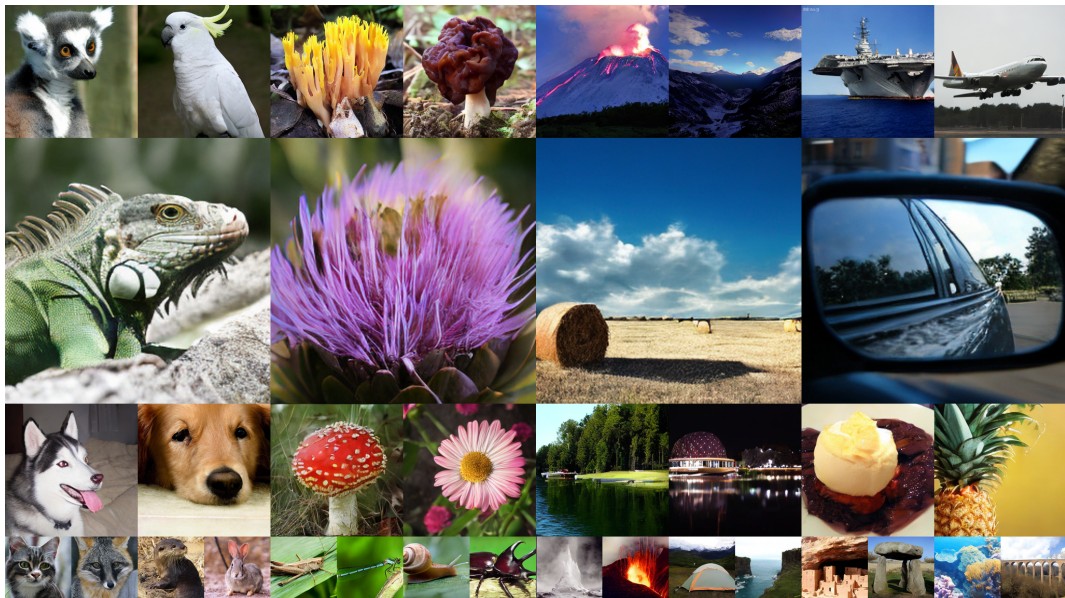

Figure 1: **RobuQ enables DiTs to generate competitive results at ultra-low bit setting.** We select $256 \times 256$ images from W1.58A3 quantized DiT-XL/2 trained on ImageNet-1K.

## Abstract

Diffusion Transformers (DiTs) have recently emerged as a powerful backbone for image generation, demonstrating superior scalability and performance over U-Net architectures. However, their practical deployment is hindered by substantial computational and memory costs. While Quantization-Aware Training (QAT) has shown promise for U-Nets, its application to DiTs faces unique challenges, primarily due to the sensitivity and distributional complexity of activations. In this work, we identify activation quantization as the primary bottleneck for pushing DiTs to extremely low-bit settings. To address this, we propose a systematic QAT framework for DiTs, named **RobuQ**. We start by establishing a strong ternary weight (W1.58A4) DiT baseline. Building upon this, we propose **RobustQuantizer** to achieve robust activation quantization. Our theoretical analyses show that the Hadamard transform can convert unknown per-token distributions into per-token normal distributions, providing a strong foundation for this method. Furthermore, we propose **AMPN**, the first **A**ctivation-only **M**ixed-**P**recision **N**etwork pipeline for DiTs. This method applies ternary weights across the entire network while allocating different activation precisions to each layer to eliminate information bottlenecks. Through extensive experiments on unconditional and conditional image generation, our RobuQ framework achieves state-of-the-art performance for DiT quantization in sub-4-bit quantization configuration. To the best of our knowledge, RobuQ is the first achieving stable and competitive image generation on large datasets like **ImageNet-1K** with activations quantized to average 2 bits.

# 1 INTRODUCTION

Recent advances in quantization-aware training (QAT) have revealed a fundamental asymmetry between weight and activation quantization in deep neural networks (Zheng et al., 2025; Feng et al., 2025a; Wang et al., 2025a; He et al., 2024b). In particular, diffusion transformer (DiT) models (Peebles & Xie, 2023), which have demonstrated strong performance in generative tasks, present unique challenges for efficient quantization due to their deep architectures and the complex distribution of activation values. While prior studies have shown that ternary quantization of weights can achieve nearly lossless accuracy (Ma et al., 2024), activation quantization remains substantially more difficult—especially for large-scale datasets like ImageNet-1K (Russakovsky et al., 2015), where the lowest reported activation bit-width is still 4 bits (Feng et al., 2025b). This highlights an opportunity to further reduce activation precision in DiT models without sacrificing generative quality greatly.

In this work, we focus on the quantization of DiT models and conduct a systematic analysis to identify activation quantization as the principal challenge in ultra-low bit settings. Building on this observation, we first establish a strong W1.58A4 DiT quantization baseline. We then theoretically demonstrate that, under our modeling assumptions, the Hadamard transform can consistently project diverse and irregular activation distributions in DiT into a standard normal form. Exploiting this property, we propose the RobustQuantizer including the construction of an advanced W1.58A4 baseline, the Hadamard transform and the robust per-token Gauss quantizer, thereby enabling highly efficient and distribution-agnostic quantization in both uniform and non-uniform quantization.

Mixed-precision quantization has recently emerged as a promising strategy to overcome the limitations of uniform ultra-low-bit quantization (Feng et al., 2025a; Zhao et al., 2024a; Kim et al., 2025; Feng et al., 2025b). We introduce the first activation-only mixed-precision quantization network (AMPN) for DiT, and explore activation bit-width allocation strategies within this framework at ultra-low bit setting. Using AMPN, we achieve SOTA image generation on ImageNet at an ultra-low precision of W1.58A3 (as seen in Fig. 1), while maintaining stable performance without collapse at the even lower bit-width of W1.58A2. Extensive experiments on both unconditional and conditional generation tasks demonstrate our method's superior performance over SOTA techniques.

Our main contributions are summarized as follows:

- Through comprehensive study, we identify activation quantization as the central bottleneck for DiTs to achieve ultra-low bit quantization. Building upon recent work, we establish a strong **baseline** for ternary weight quantization with a W1.58A4 DiT model, achieved through the integration of an SVD-initialized low-rank branch and Hadamard transform.

- We first **theoretically** demonstrate that the widely used Hadamard transform, under our modeling assumptions, can convert arbitrary activation distributions in DiT models to a per-token normal distribution. Leveraging this property, our **RobustQuantizer** supports both uniform and non-uniform quantization, achieving SOTA performance on W1.58A4.

- We introduce **AMPN**, the first DiT quantization scheme to focus exclusively on activation-only mixed-precision, and conduct a thorough exploration of activation bit-width allocation. Our method achieves SOTA performance at W1.58A3 and, furthermore, maintains stable training without collapse at an ultra-low bit setting of W1.58A2.

- Extensive evaluations across unconditional generation and conditional generation with DiT demonstrate that our quantization framework **RobuQ**, including RobustQuantizer and AMPN, consistently surpasses previous SOTA methods in both efficiency and performance, significantly advancing the feasibility of DiTs under resource constraints.

# 2 RELATED WORKS

## 2.1 DIFFUSION TRANSFORMERS

Diffusion Models (DMs) have demonstrated impressive generative capabilities across a wide range of tasks (Chen et al., 2020; Hu et al., 2022; Rombach et al., 2022; Chen et al., 2023; He et al., 2023; Li et al., 2023b;a; Liu et al., 2024; Li et al., 2024; He et al., 2024a; Ho et al., 2020; Zhao et al., 2024b; Peebles & Xie, 2023). Recent research has focused on replacing the conventional U-Net (Ronneberger et al., 2015) backbone with Transformer-based (Vaswani et al., 2017) architectures to build more powerful generative models (Croitoru et al., 2023; Rombach et al., 2022; Yang et al., 2023). Among these, Diffusion Transformers (DiTs) (Peebles & Xie, 2023) has achieved

remarkable performance in image generation, exhibiting strong scalability and significant potential for broader applications. Despite its exceptional performance, DiT still demands substantial computational resources, including high memory usage and processing power, to generate high-quality images, which significantly hinders its applicability in resource-constrained scenarios.

## 2.2 QUANTIZATION

Quantization techniques (K Esser et al., 2019; Lv et al., 2024; Zhang et al., 2024; Zhou et al., 2016) compress and accelerate neural networks by reducing the numerical precision of weights and activations (e.g., from 32-bit floating-point to 1–8-bit integers). However, applying quantization to generative tasks presents unique challenges due to the dynamic temporal nature of the diffusion process and the complex spatial structures involved (Chen et al., 2024; He et al., 2024b).

To further improve the efficiency of neural network quantization, recent research has explored even lower bit-width regimes, such as ternarization (three-value quantization) (Lu et al., 2024; Ma et al., 2024; Wang et al., 2025b) and extreme low-bit quantization (e.g., 1-bit, 2-bit) (Zheng et al., 2024; 2025). These approaches significantly reduce both memory footprint and computational complexity, but they typically struggle to maintain sufficient model expressiveness and high generation quality, especially in the context of large generative models that require intricate representations.

To address information loss caused by aggressive quantization, orthogonal transformations have been introduced into quantization pipelines (Hu et al., 2025; Lin et al., 2025; Ashkboos et al., 2024; Liu et al., 2025b). By decorrelating weights or activations before quantization (e.g., via SVD, Hadamard, or other orthogonal transforms), these methods redistribute quantization errors and better preserve information, enabling more accurate low-bit quantization for generative models.

Moreover, mixed-precision quantization has emerged as an effective strategy to balance efficiency and performance (Feng et al., 2025a; Zhao et al., 2024a; Kim et al., 2025; Feng et al., 2025b). Instead of assigning a uniform bit-width to all layers or modules, mixed-precision methods allocate higher precision to sensitive components and lower precision elsewhere, either through heuristic rules or data-driven optimization. This technique enhances quantization robustness and overall performance.

## 3 METHOD

### 3.1 ANALYSIS

Empirical evidence indicates that DiT models (Peebles & Xie, 2023) exhibit inferior performance in the low bit-width regime compared to U-Net-based (Ronneberger et al., 2015) LDM models (Rombach et al., 2022). Currently, DiT quantization is often limited to a W4A4 configuration (Liu & Zhang, 2024; Wu et al., 2024; Hwang et al., 2025; Chen et al., 2025), whereas LDM-class models have advanced to W1A4 and even W1A1 precedents (Zheng et al., 2024; 2025). This significant gap motivates a thorough investigation into activation quantization for DiTs. We identify three key challenges that hinder effective low-bit-width activation quantization in DiT models:

- **Issue 1: Lack of QAT Exploration for Ultra-low-bit Configurations.** Existing methods have primarily focused on Post-Training Quantization (PTQ) (He et al., 2024b; Wang et al., 2025a), without a thorough investigation into the boundaries of activation bit-width under the Quantization-Aware Training (QAT) framework. Compared with PTQ, QAT can explicitly optimize model parameters during training to compensate for quantization errors, thereby offering a more promising and effective route to stable ultra-low-bit deployment.

- **Issue 2: Diverse and Complex Activation Distributions.** Unlike other architectures, DiTs exhibit highly varied activation distributions across different layers and tokens (Zhao et al., 2025), posing a significant challenge due to the lack of a unified quantizer.

- **Issue 3: Potential Activation Bit-width Bottlenecks.** We find the existence of specific layers within DiT models that are particularly sensitive to activation bit-width compression, which fundamentally prevents further quantization to lower activation bit-widths.

Based on the above issues, it becomes necessary to conduct a dedicated study on ultra-low-bit activation quantization for DiT models. Our goal is to address the unique distributional and architectural challenges of DiT, and to develop strategies that maximize compression while preserving generative fidelity. Such targeted investigation is essential not only for reducing deployment costs but also for pushing the practical boundaries of DiT quantization into regimes previously thought unattainable.

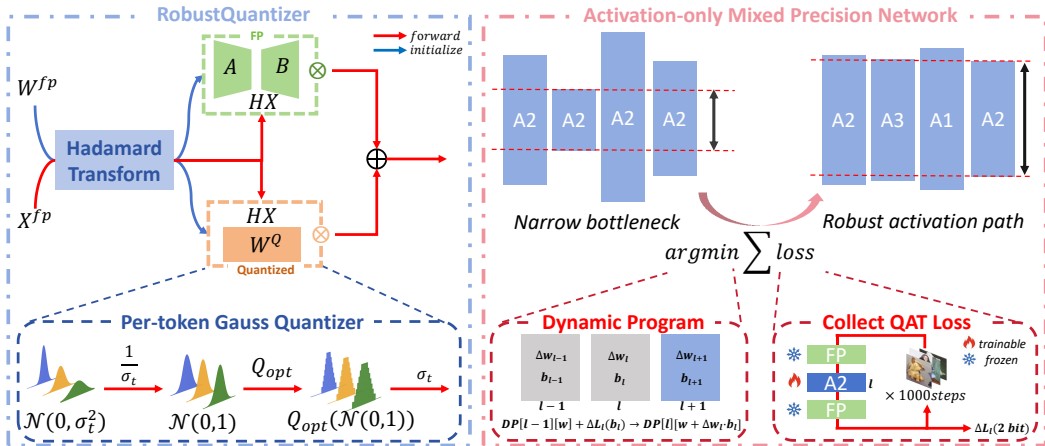

Figure 2: Overall Framework of Our Quantization Pipeline.

## 3.2 RobustQuantizer: Leveraging Hadamard Transformation Effectively

### 3.2.1 Initial Baseline and Quantization Strategy

Building upon the successful W1.58A4 configuration of BitNetv2 (Wang et al., 2025b), we establish it as our initial baseline. Specifically, we apply a Hadamard transformation (Yarlagadda & Hershey, 1993), a type of orthogonal transformation to the *proj* and *fc2* layers within the DiT modules. The Hadamard transformation is applied to both the weights and activations, where $W \leftarrow HW$ and $X \leftarrow HX$. Here, the Hadamard matrix of order $n$ is defined recursively:

$$H_1 = (1), \quad H_{2^n} = \frac{1}{\sqrt{2}} \begin{pmatrix} H_n & H_n \\ H_n & -H_n \end{pmatrix} \quad (n \geq 1). \tag{1}$$

We adopt two distinct strategies for weight and activation quantization. For weight quantization $Q_w(\cdot)$, we use a channel-wise ternarization quantizer based on the principles of BitNetV2. This maps the FP weights $W$ to discrete values per channel, as shown in the following equation:

$$Q_w(W) = \alpha \cdot \text{RoundClip}(\frac{W}{\gamma + \epsilon}, -1, 1), \tag{2}$$

where $\alpha = \text{mean}(|W|)$, $\gamma = \frac{1}{mn} \sum_{i,j} |W_{ij}|$, and $\epsilon$ is a small constant to avoid division by zero. The RoundClip function is defined as $\text{RoundClip}(x, a, b) = \min(\max(\text{round}(x), a), b)$. For activation quantization, we employ a straightforward per-token min–max quantization strategy to determine the scaling range. The quantized value $Q_x(\mathbf{x})$ for an activation tensor $\mathbf{x}$ is computed as:

$$Q_x(\mathbf{x}) = \text{clamp}\left(\left\lfloor \frac{\mathbf{x}}{\delta} \right\rfloor + \lambda, 0, 2^b - 1\right), \tag{3}$$

where $\delta = \frac{\max(\mathbf{x}) - \min(\mathbf{x})}{2^b - 1}$ is the scaling factor, $b$ is the bit-width, $\lfloor \cdot \rfloor$ denotes the floor operation, and $\lambda = -\left\lfloor \frac{\min(\mathbf{x})}{\delta} \right\rfloor$ is the zero-point that enables asymmetric quantization.

### 3.2.2 Enhanced Baseline with Integrated Techniques

Next, we turn our attention to other state-of-the-art methods. By drawing on techniques from SVD-Quant (Li et al., 2025) and BiMaCoSR (Liu et al., 2025a), we introduce a SVD-initialized low-rank matrix branch for compensation, which operates in FP. As illustrated in Fig. 2 (left), the initialization process begins with the FP weights $W$. First, a Hadamard transform is applied to $W$. Then, a truncated SVD is performed on the transformed matrix to construct the low-rank approximation, which is subsequently factorized into $A$ and $B$. The decomposition is as follows:

$$WH \approx AB = U_r \Sigma_r V_r^T. \tag{4}$$

Here, $H$ denotes the Hadamard matrix. The matrices $U_r$, $\Sigma_r$, and $V_r$ are obtained by retaining the top $r = 16$ dominant singular values and their corresponding singular vectors. The main quantized weight matrix $W^Q$ in the lower branch is also derived from the transformed weights $WH$:

$$W^Q = Q_w(WH - AB) = Q_w(W_{res}). \tag{5}$$

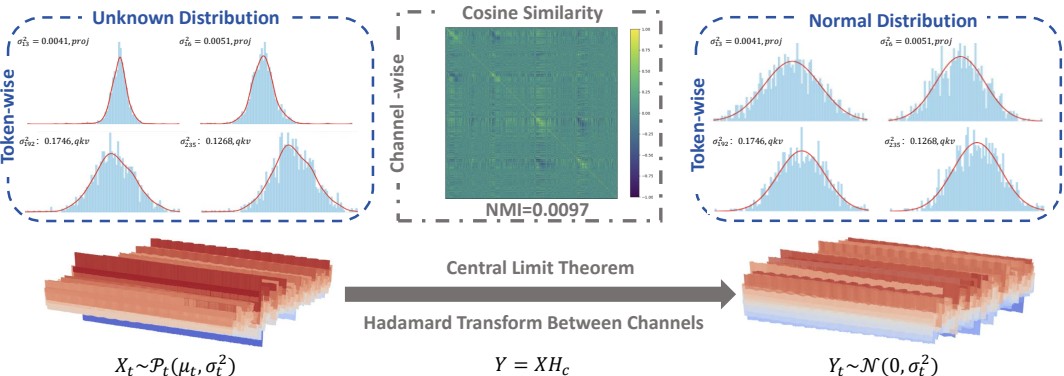

Figure 3: Illustration of how the Hadamard transforms per-token unknown distributions (left) into a known per-token normal distribution (right). Average NMI is computed across different channels.

The original weight $W$ is then approximated as follows:

$$W = WHH^\top = (AB + W_{\text{res}})H^\top \approx ABH^\top + Q_w(W_{\text{res}})H^\top. \tag{6}$$

During the forward pass, the input $X$ is passed through a Hadamard transform and then into the $Q_G$, as shown in Eq. 8. Here $Q_G$ refers to the Per-token Gauss Quantizer introduced in Section 3.2.4. The final output is the sum of the outputs from the FP low-rank branch and the main quantized branch. Although BitNetv2 restricted the Hadamard transform to *proj* and *fc2* layers, extending it to all layers, as we do here, stabilizes activation distributions and mitigates residual imbalances.

### 3.2.3 HADAMARD TRANSFORM CREATES A PER-TOKEN NORMAL DISTRIBUTION

We argue that the Hadamard transform provides more than simple activation smoothing (Kolb et al., 2023): it converts per-token activations from arbitrary distributions into predictable, approximately normal ones. This property, visualized in Fig. 3, motivates our **RobustQuantizer**.

Formally, consider the input $X \in \mathbb{R}^{T \times C}$. We have observed the following three properties:

**(i) Token-wise**: Activations across tokens within a layer share a distribution shape but differ in mean and variance, and these distributions vary significantly across layers, leading to quantization errors.

**(ii) Channel-wise**: Channels are nearly independent, with low normalized mutual information (NMI), which is a key property to satisfy the CLT assumptions (Gnedenko & Kolmogorov, 1954).

**(iii) Hadamard Matrix Property**: The normalized Hadamard matrix $H_C$ has entries of $\pm 1/\sqrt{C}$, which ensures an equal variance across the resulting transformed channels in one token.

Thus, per-token activations $X_t = (X_{t,1}, \ldots, X_{t,C})$, with $X_{t,c} \sim \mathcal{P}_{t,c}(\mu_{t,c}, \sigma_{t,c}^2)$, become

$$Y_{t,c} = \sum_{j=1}^{C} (H_C)_{j,c} X_{t,j}, \quad \text{Var}(Y_{t,c}) = \frac{1}{C} \sum_{j=1}^{C} \sigma_{t,j}^2 \triangleq \sigma_t^2. \tag{7}$$

By the Generalized CLT, $Y_t$ converges to $\mathcal{N}(0, \sigma_t^2)$, i.e., an identically distributed Gaussian for each token. This insight provides a principled theoretical foundation for achieving robust and effective per-token quantization. Further information and a formal derivation can be found in Appendix A.

### 3.2.4 FROM HADAMARD NORMALIZATION TO PER-TOKEN GAUSS QUANTIZATION

Building upon our prior analysis of how the Hadamard transform produces a per-token normal distribution, we now design the **Per-token Gauss Quantizer** $Q_G(\cdot)$ to maximally leverage this property. We present two versions of our quantizer, a **uniform** and a **non-uniform** variant. The complete process involves per token normalization using dynamically computed mean and variance and quantization with a precomputed optimal quantizer as shown in Fig. 2 (left lower). We obtain this optimal quantizer, denoted as $Q_{\text{opt}}$, by using the Lloyd-Max algorithm (Lloyd & Laboratories, 1982). The complete quantization and dequantization process for an activation vector $x$ can be expressed as:

$$x \approx Q_G(x) = \sigma_t \cdot H^T \cdot Q_{\text{opt}} \left( \frac{Hx}{\sigma_t} \right). \tag{8}$$

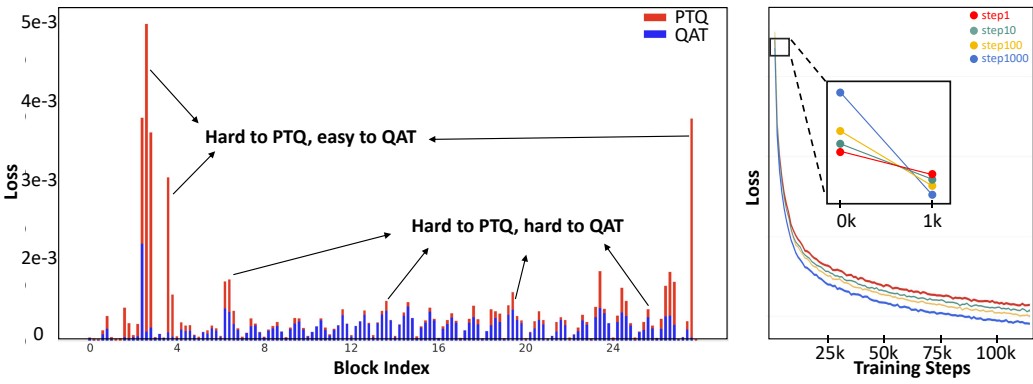

Figure 4: An illustration of why PTQ sensitivity metrics fail for ultra-low-bit QAT mixed-precision. **Left**: Visualization of accuracy loss for different linear layers with W1.58A2 quantization under PTQ and QAT (1,000 training steps). **Right**: Mixed-precision configurations derived from more QAT steps achieve a worse initial loss but a better final convergence loss.

Therefore, the forward of the quantized and low-rank FP branches can be expressed as:

$$Wx \approx \underbrace{ABHx}_{\text{FP}} + \underbrace{Q_w(W_{\text{res}}) \cdot Q_{\text{opt}}\left(\tfrac{Hx}{\sigma_t}\right) \cdot \sigma_t}_{\text{quantized}}. \tag{9}$$

### 3.3 ACTIVATION-ONLY MIXED-PRECISION NETWORK

#### 3.3.1 NAIVE PIPELINE DESIGN

We design a simple activation-only mixed-precision network (AMPN) pipeline to alleviate bottlenecks caused by uniform bit-width quantization, as shown in Fig. 2 (right). All weights are fixed to ternary (W1.58), while each activation layer $\ell \in \{1, \ldots, L\}$ selects a bit-width $b_\ell \in \mathcal{B} = \{1, 2, 3, 4\}$. The goal is to minimize accuracy loss under a target average activation bit-width $\overline{B}_{\text{tgt}}$.

To build a layer-wise sensitivity profile, we randomly sample 1,000 validation examples across timesteps and compute the mean loss gap $\Delta L_\ell(b_l)$ between the quantized and full-precision models at bit-width $b$. This metric enables a fast estimation of per-layer degradation. We then formulate bit allocation as a Dynamic Programming (DP) problem, where the objective is to minimize total loss under a resource budget. Here, $w_\ell$ is the layer-wise weight that adjusts the bitwidth contribution of each layer according to its FLOPs proportion in DiT-Block (e.g., the $w_\ell$ of mlp.fc1 is 1.334).

Among these layers, certain components are fixed for stability: the attention scores are quantized to 8 bits, and the adaLN layer to 4 bits, due to their high sensitivity yet negligible FLOPs cost (together accounting for about 2–3% of the total block computation). The optimization can be written as

$$\min_{\{b_\ell \in \mathcal{B}\}} \sum_{\ell=1}^{L} \Delta L_\ell(b_\ell) \quad \text{s.t.} \quad \frac{1}{W_{\text{tot}}} \sum_{\ell=1}^{L} w_\ell b_\ell \leq \overline{B}_{\text{tgt}}, \tag{10}$$

where $W_{\text{tot}} = \sum_{\ell=1}^{L} w_\ell$ is the total FLOPs. We solve this with DP. Let $\text{DP}[\ell][w]$ be the minimal cumulative loss after assigning bits to the first $\ell$ layers with accumulated weighted cost $w$. For the purpose of discretization, each individual layer's FLOPs and the target budget can be written as

$$\Delta w_\ell = \left\lfloor \beta \frac{w_\ell}{W_{\text{tot}}^{\text{dp}}} \right\rfloor, \qquad B = \left\lfloor \beta \overline{B}_{\text{tgt}} \right\rfloor, \tag{11}$$

where $\beta$ is a resolution factor controlling granularity (e.g. $\beta =$1,000) and $W_{\text{tot}}^{\text{dp}} = \sum_{\ell \in \mathcal{L}_{\text{dp}}} w_\ell$ is the FLOPs of layers optimized by DP. The recurrence relation is formally defined as follows,

$$\text{DP}[\ell][w + \Delta w_\ell \cdot b_\ell] = \min \left\{ \text{DP}[\ell][w + \Delta w_\ell \cdot b_\ell], \ \text{DP}[\ell - 1][w] + \Delta L_\ell(b_\ell) \right\}, \quad b_\ell \in \mathcal{B}. \tag{12}$$

We initialize $\text{DP}[0][0] = 0$ and $\text{DP}[0][w > 0] = +\infty$. The optimal cost can be formulated as

$$w^\star = \arg \min_{0 \leq w \leq B} \text{DP}[L][w], \tag{13}$$

from which the optimal allocation $\{b_\ell^\star\}$ is recovered by backtracking through the solution space.

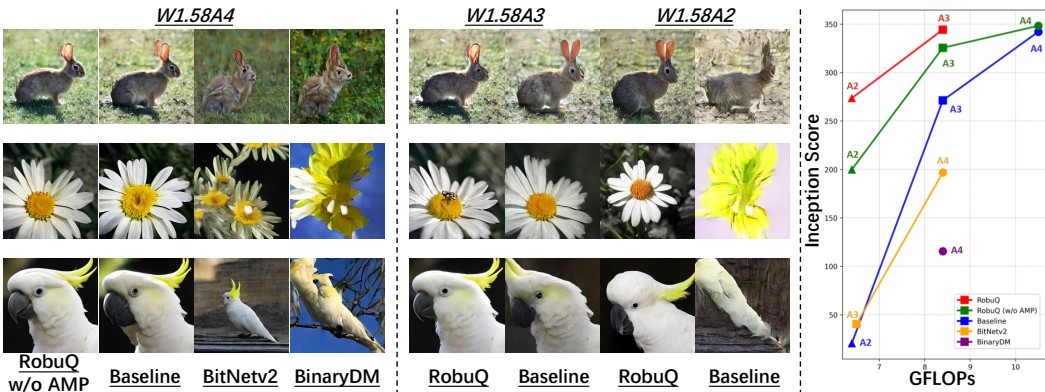

Figure 5: Visualization of the performance and efficiency of RobuQ and comparative approaches. **Left**: Our proposed RobuQ and baseline significantly outperform previous methods on W1.58A4. **Middle**: RobuQ maintains stable generation under A3 and A2 compared to collapsed baseline. **Right**: The RobuQ series achieve higher Inception Scores under the same FLOPs.

### 3.3.2 ULTRA-LOW-BIT QAT

Mixed-precision methods (Feng et al., 2025a; Zhao et al., 2024a; Kim et al., 2025; Feng et al., 2025b) have traditionally employed PTQ to collect parameters, as they are often applied in mid-bit configurations. However, our work targets ultra-low-bit quantization under the QAT framework. In this setting, even if a layer exhibits large quantization errors during PTQ, the model can still compensate for these errors during training, making QAT more adaptable, as shown in Fig. 4 (left). On the other hand, a low quantization error observed in PTQ does not necessarily ensure a consistent or further reduction in quantization errors during the subsequent QAT process.

To investigate this, we explored training the quantized layers for different numbers of steps while collecting quantization errors. Specifically, we trained for 1, 10, 100, and 1,000 steps, using the same learning rate as standard training. Our findings revealed that while schemes with fewer training steps (such as 1 and 10 steps) initially exhibited lower quantization errors, those trained with more steps (such as 1,000 steps) achieved a significantly lower final convergence loss, as shown in Fig. 4 (right). This aligns with our hypothesis that additional QAT steps allow the model to better adjust to the quantization process, gradually improving its performance and robustness over time.

## 4 EXPERIMENTS

### 4.1 SETUP

**Datasets and Evaluation Metrics**. We evaluate pre-trained class-conditional DiT-XL/2 models at $256 \times 256$ resolution on ImageNet-1K (Russakovsky et al., 2015) and FFHQ (Karras et al., 2019). The DDPM solver (Ho et al., 2020) with 250 sampling steps is employed for the generation process. For all methods under evaluation, we uniformly sample a total of 10,000 generated images for both the ImageNet-1K $256 \times 256$ and FFHQ $256 \times 256$ benchmarks. We use four metrics to assess generated image quality: Fréchet Inception Distance (FID) (Heusel et al., 2017), spatial FID (sFID) (Salimans et al., 2016; Nash et al., 2021), Inception Score (IS) (Salimans et al., 2016; Barratt & Rharma, 2018), and Precision, all computed using the ADM toolkit (Dhariwal & Nichol, 2021).

**Compared Methods**. We compare our RobuQ series (where RobuQ (w/o AMP) denotes using only the RobustQuantizer) with SOTA quantization approaches, covering both PTQ and QAT paradigms. These include BitNetv2 (Wang et al., 2025b) and QueST (Wang et al., 2025a) for ultra-low-bit QAT, PTQ4DiT (Wu et al., 2024) and Q-DiT (Chen et al., 2025) as DiT-specific PTQ methods, and BinaryDM (Zheng et al., 2025) for QAT binarized diffusion models. We also incorporate Quarot (Ashkboos et al., 2024) and SVD-Quant (Li et al., 2025) as components of our strong baseline.

**Training and Quantization Details.** All experiments are conducted with PyTorch (Paszke et al., 2019) on a single NVIDIA RTX A6000-48GB GPU. For all QAT methods, we use the AdamW optimizer (Loshchilov & Hutter, 2019) (learning rate=$10^{-5}$, weight decay=0) with a batch size of 8 and train for 350k iterations. We keep the embedding and final layer in full precision across all methods, and maintain 8-bit precision for activation-activation matrix multiplication operations, as they constitute a small fraction of the computation and exhibit high sensitivity to quantization.

Table 1: Performance on ImageNet-1K 256×256 and FFHQ 256×256 under different settings.

| Setting | Method | Bit-width (W/A) | IS↑ | FID↓ | sFID↓ | Precision↑ |
|---|---|---|---|---|---|---|
| ImageNet steps= 50 cfg = 1.5 | FP | 32/32 | 239.50 | 6.62 | 21.10 | 0.7849 |
| | QueST | 4/4 | 4.87 | 215.06 | 72.15 | 0.0529 |
| | PTQ4DiT | 4/4 | 3.05 | 231.80 | 106.42 | 0.1003 |
| | Q-DiT | 4/4 | 2.01 | 248.11 | 404.44 | 0.0138 |
| | BinaryDM† | 1.58/4 | 25.63 | 62.91 | 38.28 | 0.3765 |
| | Bitnetv2 | 1.58/4 | 44.32 | 41.59 | 34.09 | 0.5002 |
| | Baseline | 1.58/4 | 95.07 | 20.82 | 27.53 | 0.6152 |
| | **RobuQ (w/o AMP)** | 1.58/4 | **103.24** | **17.97** | **26.95** | **0.6577** |
| | Baseline | 1.58/3 | 51.31 | 40.23 | 35.64 | 0.4946 |
| | **RobuQ (w/o AMP)** | 1.58/3 | 83.84 | 24.44 | 29.18 | 0.6001 |
| | **RobuQ** | 1.58/3 | **93.75** | **21.40** | **26.99** | **0.6190** |
| | Baseline | 1.58/2 | 10.63 | 120.49 | 62.29 | 0.2091 |
| | **RobuQ (w/o AMP)** | 1.58/2 | 45.65 | 43.31 | 38.89 | 0.4917 |
| | **RobuQ** | 1.58/2 | **66.74** | **30.30** | **30.66** | **0.5680** |
| ImageNet steps= 50 cfg = 4.0 | FP | 32/32 | 478.35 | 19.11 | 21.61 | 0.9298 |
| | QueST | 4/4 | 42.07 | 84.20 | 45.67 | 0.2651 |
| | PTQ4DiT | 4/4 | 5.64 | 144.07 | 85.83 | 0.1078 |
| | Q-DiT | 4/4 | 8.23 | 141.13 | 279.26 | 0.1272 |
| | BinaryDM† | 1.58/4 | 115.52 | 17.08 | 23.15 | 0.7230 |
| | Bitnetv2 | 1.58/4 | 196.78 | **11.69** | 21.44 | 0.8370 |
| | Baseline | 1.58/4 | 342.07 | 12.82 | 20.05 | 0.9092 |
| | **RobuQ (w/o AMP)** | 1.58/4 | **349.22** | 12.64 | **19.69** | **0.9186** |
| | Baseline | 1.58/3 | 254.92 | **10.83** | 21.68 | 0.8585 |
| | **RobuQ (w/o AMP)** | 1.58/3 | 325.56 | 12.31 | 19.94 | 0.9053 |
| | **RobuQ** | 1.58/3 | **342.94** | 12.71 | **19.87** | **0.9129** |
| | Baseline | 1.58/2 | 20.65 | 75.02 | 35.81 | 0.2812 |
| | **RobuQ (w/o AMP)** | 1.58/2 | 200.00 | 11.97 | 21.73 | 0.8188 |
| | **RobuQ** | 1.58/2 | **273.58** | **11.06** | **21.57** | **0.8751** |
| ImageNet steps= 150 cfg = 4.0 | FP | 32/32 | 479.72 | 19.67 | 22.94 | 0.9301 |
| | QueST | 4/4 | 49.55 | 76.55 | 43.26 | 0.2876 |
| | PTQ4DiT | 4/4 | 5.07 | 148.17 | 95.02 | 0.0982 |
| | Q-DiT | 4/4 | 10.74 | 124.48 | 286.99 | 0.1598 |
| | BinaryDM† | 1.58/4 | 124.78 | 15.50 | 21.66 | 0.7575 |
| | Bitnetv2 | 1.58/4 | 206.28 | **11.72** | 20.62 | 0.8591 |
| | Baseline | 1.58/4 | 344.43 | 13.92 | 20.75 | 0.9167 |
| | **RobuQ (w/o AMP)** | 1.58/4 | **348.40** | 13.82 | **20.32** | **0.9225** |
| | Baseline | 1.58/3 | 271.27 | **11.55** | 20.76 | 0.8876 |
| | **RobuQ (w/o AMP)** | 1.58/3 | 333.62 | 13.65 | 20.75 | 0.9180 |
| | **RobuQ** | 1.58/3 | **342.73** | 14.27 | **20.63** | **0.9247** |
| | Baseline | 1.58/2 | 23.22 | 68.04 | 30.84 | 0.2999 |
| | **RobuQ (w/o AMP)** | 1.58/2 | 220.47 | **11.17** | **20.10** | 0.8573 |
| | **RobuQ** | 1.58/2 | **281.73** | 11.86 | 21.94 | **0.8922** |
| FFHQ steps= 50 Uncondition | FP | 32/32 | N/A | 11.71 | 28.88 | 0.7526 |
| | QueST | 4/4 | N/A | 72.88 | 85.56 | 0.1897 |
| | Bitnetv2 | 1.58/4 | N/A | 66.55 | 64.49 | 0.3499 |
| | Baseline | 1.58/4 | N/A | 34.32 | 44.37 | 0.5771 |
| | **RobuQ (w/o AMP)** | 1.58/4 | N/A | **25.62** | **37.15** | **0.6228** |
| | Baseline | 1.58/3 | N/A | 35.11 | 43.07 | 0.5988 |
| | **RobuQ** | 1.58/3 | N/A | **28.11** | **38.43** | **0.6128** |
| | Baseline | 1.58/2 | N/A | 59.50 | 62.58 | 0.4159 |
| | **RobuQ** | 1.58/2 | N/A | **38.13** | **42.29** | **0.5568** |

† For fairness, we swapped BinaryDM's binarization for ternarization.

## 4.2 MAIN RESULT

As shown in Table 1, on both ImageNet-1K 256 × 256 with low classifier-free guidance (Ho & Salimans, 2022) (cfg=1.5, 50 steps) and FFHQ 256×256 under unconditional generation (50 steps), our quantized models demonstrate comprehensive superiority across all bit-widths, which solidly validates the effectiveness of our approach. However, when employing a higher guidance scale (cfg=4.0), although our method achieves better metrics, FID exhibits anomalous behavior: all quantized methods surprisingly outperform the FP model, showing an inverse relationship with other metrics. This phenomenon suggests the need for more precise evaluation metrics in low-bit setting. Increasing sampling steps from 50 to 150 maintains consistent trends. Fig. 5 (left and middle) provides visually comparative results across diverse methods and different bit-widths.

Table 2: Ablation studies on ImageNet-1K 256×256. Timesteps are 50 and cfg is 1.5.

| Method | IS↑ | FID↓ | sFID↓ | Precision↑ |
|---|---|---|---|---|
| BitNetv2 | 44.32 | 41.59 | 34.09 | 0.5002 |
| + LRB | 68.82 | 29.59 | 31.35 | 0.5807 |
| **+ LRB + All Hadamard** | **95.07** | **20.82** | **27.53** | **0.6152** |

(a) Baseline construction at W1.58A4.

| Method | IS↑ | FID↓ | sFID↓ | Precision↑ |
|---|---|---|---|---|
| Baseline | 95.07 | 20.82 | 27.53 | 0.6152 |
| +Non-uniform Quantizer | 96.19 | 20.33 | 27.52 | 0.6262 |
| **+Uniform Quantizer** | **103.24** | **17.97** | **26.95** | **0.6577** |

(b) Per-token Gauss Quantizer at W1.58A4.

| QAT-step | Method | IS↑ | FID↓ | sFID↓ | Precision↑ | Training-time↓ |
|---|---|---|---|---|---|---|
| N/A | RobustQuantizer | 45.65 | 43.31 | 38.89 | 0.4917 | 126.0h |
| 1 | | 50.67 | 41.06 | 34.58 | 0.5028 | 3.1h+126.0h |
| 10 | | 52.12 | 39.01 | 32.50 | 0.5092 | 3.7h+126.0h |
| 100 | +AMP | 56.31 | 37.45 | 32.61 | 0.5097 | 9.5h+126.0h |
| 500 | | 57.93 | 36.57 | 33.46 | 0.5213 | 36.0h+126.0h |
| 1,000 | | 66.74 | 30.30 | 30.66 | 0.5680 | 78.5h+126.0h |
| 1,500 | | 66.23 | 30.56 | 30.23 | 0.5701 | 121.0h+126.0h |

(c) AMPN at W1.58A2. Training-time comprises metric collection and actual training.

Table 3: Inference efficiency of our proposed RobuQ of DiT-XL/2 on ImageNet-1K 256×256.

| cfg=4.0 step=50 | FP | QueST | BinaryDM | Bitnetv2 | Baseline | RobuQ | RobuQ | RobuQ |
|---|---|---|---|---|---|---|---|---|
| W/A | 32/32 | 4/4 | 1.58/4 | 1.58/4 | 1.58/4 | 1.58/4 | 1.58/3 | 1.58/2 |
| Size (MB)↓ | 2,575.4 | 341.22 | 148.13 | 148.13 | 194.75 | 194.75 | 194.75 | 194.75 |
| FLOPs (G)↓ | 114.52 | 14.94 | 8.04 | 8.04 | 10.07 | 10.07 | 8.34 | 6.61 |
| IS↑ / Precision↑ | 478.35/0.9298 | 42.07/0.2651 | 115.52/0.7230 | 196.78/0.8370 | 342.07/0.9092 | 349.22/0.9186 | 344.29/0.9083 | 273.58/0.8751 |

## 4.3 ABLATION STUDY

**Baseline Construction.** We first conducted ablation studies on our baseline components, as shown in Table 2a. Starting with BitNetv2, adding a low-rank matrix branch (LRB) significantly improves performance and accelerates convergence, with the FID dropping from 41.59 to 29.59. Applying a full Hadamard transformation to all linear layers in DiT pushes the FID even lower to 20.82, demonstrating the crucial role of both components in establishing a robust foundation for our method.

**Per-token Gauss Quantizer.** As shown in Table 2b, while the theoretically optimal non-uniform quantizer only provides a slight performance gain with its FID dropping from 20.82 to 20.33, uniform quantizer achieves a much lower FID of 17.97 and a higher IS of 103.24. This is because the uniform quantizer is more robust to the small approximation errors inherent in real-world activations, proving to be more effective and stable in practice. Therefore, considering both the performance and the ease of deployment, we have adopted the uniform quantizer as final choice.

**QAT Steps in AMP.** As shown in Table 2c, we determined the optimal number of QAT steps to collect metrics in our AMP method. Performance consistently improved with an increasing number of QAT steps. For example, the FID dropped from 41.06 at one step to 39.01 at 10 steps, and then further to 30.30 at 1,000 steps. The optimal balance between quality and cost was found at 1,000 steps, which yielded the best FID of 30.30. Increasing the steps to 1,500 offered only negligible gains at a significant additional cost. Therefore, we adopted the 1,000-step approach for all evaluations.

## 4.4 EFFICIENCY ANALYSIS

Table 3 demonstrates that the RobuQ series achieves the best efficiency-accuracy trade-off. Notably, our RobuQ at W1.58A2 even surpasses BinaryDM and Bitnetv2 at W1.58A4, while providing a 17.3× theoretical speedup ratio and a 13.2× model compression ratio compared to the FP model. The visual comparisons in Fig. 5 (right) further corroborate these results. More details including the model FLOPs breakdown and computation are provided in Appendix C.

## 5 CONCLUSION

We revisit quantization for Diffusion Transformers and identify the activation pathway as the primary bottleneck for ultra-low-bit deployment. Building on a strong W1.58A4 baseline featuring an SVD-initialized low-rank branch and all-layer Hadamard mixing, we demonstrate that the Hadamard transform effectively Gaussianizes per-token activations. This enables the development of a distribution-agnostic RobustQuantizer. Its hardware-friendly uniform implementation, when integrated with an activation-only mixed-precision network (AMPN), achieves stable training and delivers significant quality improvements. Together, these advancements establish new state-of-the-art results for quantized DiTs, ultimately pushing their capabilities to the W1.58A2 configuration.

## ETHICS STATEMENT

This work focuses on developing quantization-aware training techniques for Diffusion Transformers to improve efficiency and scalability. The research does not involve human or animal subjects, personal or sensitive data, or any proprietary datasets. All datasets used in the experiments, including ImageNet-1K and FFHQ, are publicly available and widely used within the research community. Our contributions are strictly methodological and intended to advance efficient and responsible AI research by reducing the computational and memory requirements of large generative models. We confirm that this study complies fully with the ICLR Code of Ethics.

## REPRODUCIBILITY STATEMENT

We have made extensive efforts to ensure the reproducibility of our work. All code implementations used in this paper will be released publicly to facilitate independent verification and further research. Complete theoretical derivations are provided in Appendix A, where we present formal proofs of the Hadamard transform producing approximately normal per-token coordinates. The detailed algorithmic pipeline of our Activation Mixed-Precision Network (AMPN) is included in Appendix B, offering a comprehensive description of the quantization and training procedures. Experimental configurations, including datasets, training schedules, and evaluation protocols, are reported in Section 4.1. Together, these resources ensure that our results can be faithfully reproduced.

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

# A  FORMAL PROOF: HADAMARD TRANSFORM PRODUCES APPROXIMATELY NORMAL PER-TOKEN COORDINATES

**Notation aligned with the main text.** Let a single token's activation vector be denoted by $\mathbf{x} \in \mathbb{R}^C$ (token index $t$ suppressed for clarity). We use the normalized Hadamard transform (Yarlagadda & Hershey, 1993)

$$H \in \left\{\pm \tfrac{1}{\sqrt{C}}\right\}^{C \times C}, \qquad H^\top H = HH^\top = I_C, \tag{14}$$

and define the transformed coordinates $\mathbf{y} = H\mathbf{x}$. For channel $j$, set

$$\mu_j = \mathbb{E}[x_j], \quad \tilde{x}_j := x_j - \mu_j, \quad \sigma_j^2 = \mathrm{Var}(x_j), \tag{15}$$

and define the per-token average variance

$$\sigma_t^2 := \frac{1}{C} \sum_{j=1}^C \sigma_j^2. \tag{16}$$

We also write the Hadamard coefficients as $a_j^{(c)} := H_{cj} = \pm\tfrac{1}{\sqrt{C}}$, so that

$$y_c = \sum_{j=1}^C a_j^{(c)} x_j, \qquad c = 1, \ldots, C. \tag{17}$$

## A.1  EXACT SECOND-MOMENT IDENTITIES

By linearity and orthogonality (distribution-free), for each coordinate $c$,

$$\mathbb{E}[y_c] = \sum_{j=1}^C a_j^{(c)} \mu_j, \tag{18}$$

$$\mathrm{Var}(y_c) = \sum_{j=1}^C (a_j^{(c)})^2 \sigma_j^2 = \frac{1}{C} \sum_{j=1}^C \sigma_j^2 = \sigma_t^2, \tag{19}$$

$$\mathrm{Cov}(y_c, y_{c'}) = \sum_{j=1}^C a_j^{(c)} a_j^{(c')} \sigma_j^2 = \frac{1}{C} \sum_{j=1}^C s_j^{(c,c')} \sigma_j^2, \quad s_j^{(c,c')} := \mathrm{sign}(H_{cj} H_{c'j}) \in \{\pm 1\}. \tag{20}$$

Equation 19 shows exact variance equalization across transformed channels; equation 20 expresses off-diagonals as signed averages of per-channel variances.

## A.2  CENTRAL LIMIT THEOREM AND ASYMPTOTIC INDEPENDENCE

**Assumptions (A1–A3).**

- (A1) The centered variables $\tilde{x}_j$ are independent (or weakly dependent in a manner admitting triangular-array CLTs (Gnedenko & Kolmogorov, 1954)).

- (A2) There exists $\kappa > 0$ with $\sup_j \mathbb{E}[|\tilde{x}_j|^{2+\kappa}] < \infty$; in particular $\sup_j \mathbb{E}[|\tilde{x}_j|^3] \le M_3 < \infty$.

- (A3) No adversarial alignment of $\{\mu_j\}, \{\sigma_j^2\}$ with Hadamard sign patterns (practically, variance deviations are not aligned with a single Hadamard row/column).

**Univariate CLT (Berry–Esseen).** Fix $c$ and consider the triangular-array terms $\xi_j^{(C)} := a_j^{(c)} \tilde{x}_j$. Their variance sum is

$$s_C^2 = \sum_{j=1}^C \mathrm{Var}(\xi_j^{(C)}) = \sum_{j=1}^C (a_j^{(c)})^2 \sigma_j^2 = \sigma_t^2. \tag{21}$$

Because $|a_j^{(c)}| = 1/\sqrt{C}$,

$$\sum_{j=1}^{C} \mathbb{E}[|\xi_j^{(C)}|^3] \;=\; \frac{1}{C^{3/2}} \sum_{j=1}^{C} \mathbb{E}[|\tilde{x}_j|^3] \;\leq\; \frac{M_3}{\sqrt{C}}. \tag{22}$$

Berry–Esseen for non-identical summands yields an absolute constant $K_{\mathrm{BE}}$ such that

$$\sup_{x\in\mathbb{R}} \Big| \Pr\Big( \frac{\sum_{j=1}^{C} \xi_j^{(C)}}{\sigma_t} \leq x \Big) - \Phi(x) \Big| \;\leq\; \frac{K_{\mathrm{BE}}\, M_3}{\sigma_t^3 \sqrt{C}}. \tag{23}$$

Thus each scalar coordinate (after centering and normalization) converges to $\mathcal{N}(0,1)$ with Kolmogorov error $O(C^{-1/2})$ (Bobkov, 2023; Bentkus, 1997).

**Finite-Dimensional Gaussian Convergence.**  For fixed indices $c_1,\dots,c_m$ (with $m$ independent of $C$) and any $\lambda \in \mathbb{R}^m$,

$$L_C(\lambda) \;:=\; \sum_{r=1}^{m} \lambda_r \frac{\sum_{j=1}^{C} a_j^{(c_r)} \tilde{x}_j}{\sigma_t} \;=\; \frac{1}{\sigma_t} \sum_{j=1}^{C} \Big( \sum_{r=1}^{m} \lambda_r a_j^{(c_r)} \Big) \tilde{x}_j, \tag{24}$$

where the inner coefficients are $O(C^{-1/2})$ uniformly in $j$. Standard Lyapunov/Lindeberg conditions hold, implying

$$L_C(\lambda) \xrightarrow{d} \mathcal{N}\big(0,\; \lambda^\top \Lambda \lambda\big), \tag{25}$$

with limit covariance $\Lambda$ determined by equation 20. By Cramér–Wold, $(y_{c_1},\dots,y_{c_m})$ converges to a multivariate Gaussian whose diagonal entries equal $\sigma_t^2$ (Billingsley, 1995; Lyons & Zumbrun, 2017).

## A.3 Off-Diagonal Covariance Decay and Asymptotic Independence

Write variance deviations $\delta_j := \sigma_j^2 - \sigma_t^2$. From equation 20,

$$\mathrm{Cov}(y_c, y_{c'}) \;=\; \frac{1}{C} \sum_{j=1}^{C} s_j^{(c,c')} \delta_j. \tag{26}$$

Orthogonality of Hadamard rows implies near-cancellation of the $\pm 1$ signs in the average; two practical sufficient conditions ensuring $\mathrm{Cov}(y_c, y_{c'}) \to 0$ as $C \to \infty$ are:

- *Uniform small deviations:* $\max_j |\delta_j| \to 0 \Rightarrow |\mathrm{Cov}(y_c, y_{c'})| \leq \max_j |\delta_j| \to 0$.

- $\ell_2$-small deviations: letting $\mathbf{w}^{(c,c')} = (s_1^{(c,c')}, \dots, s_C^{(c,c')})$,

$$|\mathrm{Cov}(y_c, y_{c'})| = \frac{1}{C} \big| \langle \boldsymbol{\delta}, \mathbf{w}^{(c,c')} \rangle \big| \leq \frac{\|\boldsymbol{\delta}\|_2}{\sqrt{C}}, \quad \Rightarrow \|\boldsymbol{\delta}\|_2 = o(\sqrt{C}) \implies \mathrm{Cov}(y_c, y_{c'}) \to 0. \tag{27}$$

Combined with the finite-dimensional CLT, this yields asymptotic joint Gaussianity with diagonal covariance $\sigma_t^2 I_m$; hence the transformed coordinates become asymptotically independent Gaussians.

## A.4 Quantitative Closeness to a Product Gaussian: KL and TV Bounds

Let $\Sigma_m$ be the covariance of $(y_{c_1},\dots,y_{c_m})$ and decompose

$$\Sigma_m \;=\; \sigma_t^2 I_m + E_m, \tag{28}$$

where $E_m$ has zeros on the diagonal and off-diagonals $e_{ij} = \mathrm{Cov}(y_{c_i}, y_{c_j})$. Then

$$\mathrm{KL}\big( \mathcal{N}(\mu_m, \Sigma_m) \,\big\|\, \mathcal{N}(\mu_m, \sigma_t^2 I_m) \big) = -\tfrac{1}{2} \ln \det\big( I_m + \sigma_t^{-2} E_m \big). \tag{29}$$

If $\|\sigma_t^{-2} E_m\|_{\mathrm{op}} < \tfrac{1}{2}$ (Petersen & Pedersen, 2012), expand $\ln \det(I + A)$ to obtain

$$\mathrm{KL} \;=\; \tfrac{1}{4} \sigma_t^{-4} \|E_m\|_F^2 \;+\; O\big( \|E_m\|_F^3 / \sigma_t^6 \big). \tag{30}$$

Using $\|E_m\|_F^2 \le m(m-1)\max_{i \ne j} e_{ij}^2$ and $|e_{ij}| \le \|\boldsymbol{\delta}\|_2/\sqrt{C}$ from Section A.3,

$$\text{KL} = O\left(\frac{m^2\|\boldsymbol{\delta}\|_2^2}{C\,\sigma_t^4}\right). \tag{31}$$

By Pinsker (Csiszár & Körner, 2011; Canonne, 2022), $\text{TV} \le \sqrt{\frac{1}{2}\text{KL}}$, hence

$$\text{TV}\big(\mathcal{N}(\mu_m, \Sigma_m),\, \mathcal{N}(\mu_m, \sigma_t^2 I_m)\big) = O\left(\frac{m\,\|\boldsymbol{\delta}\|_2}{\sqrt{C}\,\sigma_t^2}\right). \tag{32}$$

The total deviation of the true law of $(y_{c_1}, \ldots, y_{c_m})$ from a product Gaussian equals the multivariate non-Gaussianity error (Berry–Esseen/Bentkus type, $O(C^{-1/2})$) plus equation 32. Thus, for fixed $m$,

$$\text{TV}_{\text{total}} = O\big(C^{-1/2}\big) + O\left(\frac{m\,\|\boldsymbol{\delta}\|_2}{\sqrt{C}\,\sigma_t^2}\right), \tag{33}$$

which vanishes at rate $O(C^{-1/2})$ when $\|\boldsymbol{\delta}\|_2 = o(\sqrt{C})$.

### A.5 QUANTIZATION AND MEAN-SQUARED ERROR

Under the mean-squared error (MSE) metric, applying the Hadamard transform for quantization does not change the final quantization error. This conclusion follows from the orthogonality of the Hadamard matrix (after normalization).

Let the activation vector be $X$, and the transformed vector be $Y = HX$. If we quantize $Y$ to get $Q(Y)$ and then recover the vector via the inverse transform, the resulting vector is $X_{\text{rec}} = H^\top Q(Y) = H^\top Q(HX)$.

The MSE of the quantization error is:

$$\text{MSE} = \mathbb{E}[\|X - X_{\text{rec}}\|_2^2] = \mathbb{E}[\|X - H^\top Q(HX)\|_2^2] \tag{34}$$

Since an orthogonal transform preserves the Euclidean norm (length) of a vector, we have:

$$\text{MSE} = \mathbb{E}[\|H(X - H^\top Q(HX))\|_2^2] = \mathbb{E}[\|HX - HH^\top Q(HX)\|_2^2] = \mathbb{E}[\|HX - Q(HX)\|_2^2] \tag{35}$$

This shows that $\text{MSE} = \mathbb{E}[\|Y - Q(Y)\|_2^2]$. This identity demonstrates that the mean-squared error of quantizing the original vector $X$ is identical to the mean-squared error of quantizing the transformed vector $Y$. This means our objective can shift from "how to quantize $X$" to "how to quantize $Y$."

As proven in this document, the coordinates of $Y$ are approximately Gaussian and nearly independent. This provides a great convenience for designing a quantizer. We can now transform a complex multivariate quantization problem into quantizing a series of approximately independent Gaussian variables.

The Gaussian distribution has the highest entropy among all continuous distributions with a given variance (Cover & Thomas, 2006; Jaynes, 1957). From an information-theoretic perspective, this means it contains the maximum randomness or "uncertainty." Therefore, for a given number of quantization bits, quantizing a Gaussian distribution is the "most difficult" task and typically results in the largest quantization error. Our method effectively prepares for this "worst-case" scenario.

By designing a quantizer optimized for the Gaussian distribution, we ensure that the quantization scheme is robust and effective for the Hadamard-transformed activations under the MSE metric.

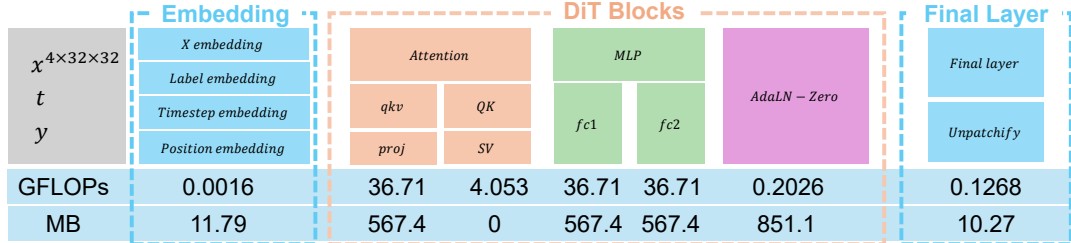

Figure 6: FLOPs and Memory Breakdown in DiT-XL/2 Model.

---

**Fundamental Insight: Distribution-Agnostic Quantization via Whitening** 💡

**Core Idea:** We note that a lot existing quantization works focus on observing data distributions to extract prior knowledge and design corresponding quantizers, but what if we *erase* prior knowledge instead?

**The RobustQuantizer Paradigm:**

- No Prior Assumptions: Instead of modeling input statistics, we use random orthogonal projections to *actively transform* inputs into the worst-case distribution – Gaussian noise

- Embracing the Hardest Case: While $\mathcal{N}(0,1)$ has minimal information (maximum entropy), its perfect knownness allows pre-computing optimal quantization parameters

- Theoretical Guarantee: This establishes a rigorous *lower-bound* for quantization performance *without requiring any prior knowledge* about input distributions

**Future Work:**

> Exploring alternative transformation methods to *other known distributions* and investigating their trade-offs between information preservation and quantization efficiency will be our key focus for future work.

---

## B ACTIVATION MIXED PRECISION NETWORK PIPELINE

We summarize the AMPN pipeline succinctly and provide compact pseudocode for the major algorithmic components. Here, $L$ is the number of activation layers, $b_\ell \in \mathcal{B} = \{1, 2, 3, 4\}$ is the chosen activation bit-width for layer $\ell$, weights are frozen to ternary (W1.58), and $w_\ell$ denotes the FLOPs-based cost weight for layer $\ell$ with $W_{\text{tot}} = \sum_\ell w_\ell$. For a single-layer QAT run let $\Delta L_\ell(b_l) = L_{\ell,b_l} - L_{\text{FP}}$ be the validation loss gap after short training; the constrained objective is

$$\min_{\{b_\ell\}} \sum_{\ell=1}^{L} \Delta L_\ell(b_\ell) \quad \text{s.t.} \quad \frac{1}{W_{\text{tot}}} \sum_{\ell=1}^{L} w_\ell b_\ell \leq \overline{B}_{\text{tgt}}. \tag{36}$$

The pipeline is as follows:

1. Run Algorithm 1 to obtain the QAT-based sensitivity table $\Delta L_\ell(b_l)$. This involves briefly training each layer individually at a given bit-width while all other layers are frozen in FP.

2. Generate the optimal bit-width allocation $C_{\text{dp}}$ by running Algorithm 2 with the sensitivity table from the previous step.

3. Train the selected allocation $C_{\text{dp}}$ end-to-end with a full QAT schedule.

## C EFFICIENCY ANALYSIS AND DEPLOYMENT

### C.1 FLOPS AND MEMORY BREAKDOWN IN DIT-XL/2 MODEL

Here we provide an analysis of the FLOPs and memory usage for the DiT-XL/2 model (Peebles & Xie, 2023), as shown in Figure 6. As illustrated, the DiT block accounts for the vast majority of

---

**Algorithm 1** QAT-based Sensitivity Profiling

---

**Require:** FP model $\mathcal{M}$, validation pool $\mathcal{V}_{\text{pool}}$, training data $\mathcal{D}_{\text{train}}$, bit set $\mathcal{B}$, short QAT steps $T_{\text{short}}$
**Ensure:** sensitivity table $\Delta L_\ell(b_l)$ for all $\ell, b_l$
1: $L_{\text{FP}} \leftarrow \text{Eval}(\mathcal{M}, \mathcal{V}_{\text{pool}})$
2: **for** each layer $\ell = 1, \ldots, L$ **do**
3:     **for** each $b_l \in \mathcal{B}$ **do**
4:         $\mathcal{M}_{\text{q}} \leftarrow \text{Copy}(\mathcal{M})$
5:         Quantize layer $\ell$ of $\mathcal{M}_{\text{q}}$ to $b_l$ bits.
6:         Freeze parameters of all layers $\ell' \neq \ell$ in $\mathcal{M}_{\text{q}}$.
7:         Train $\mathcal{M}_{\text{q}}$ for $T_{\text{short}}$ steps on $\mathcal{D}_{\text{train}}$.
8:         $L_{\ell,b_l} \leftarrow \text{Eval}(\mathcal{M}_{\text{q}}, \mathcal{V}_{\text{pool}})$.
9:         $\Delta L_\ell(b_l) \leftarrow L_{\ell,b_l} - L_{\text{FP}}$.
10:    **end for**
11: **end for**
12: **return** $\Delta L_\ell(b_l)$

---

**Algorithm 2** Discretized DP for Bit-Width Allocation

---

**Require:** sensitivity table $\Delta L_\ell(b_l)$ for $\ell \in \mathcal{L}_{\text{dp}}$, layer costs $w_\ell$, resolution $\beta$, target $\overline{B}_{\text{tgt}}$
**Ensure:** DP-optimal allocation $C_{\text{dp}}$ on $\mathcal{L}_{\text{dp}}$
1: Compute total DP cost $W_{\text{tot}}^{\text{dp}} \leftarrow \sum_{\ell \in \mathcal{L}_{\text{dp}}} w_\ell$
2: **for** each $\ell \in \mathcal{L}_{\text{dp}}$ **do**
3:     Discretize layer cost: $\Delta w_\ell \leftarrow \left\lfloor \beta \frac{w_\ell}{W_{\text{tot}}^{\text{dp}}} \right\rfloor$
4: **end for**
5: Discretize target budget: $B \leftarrow \left\lfloor \beta \overline{B}_{\text{tgt}} \right\rfloor$
6: Initialize $\text{DP}[0 \ldots |\mathcal{L}_{\text{dp}}|][0 \ldots B] \leftarrow +\infty$; $\text{DP}[0][0] \leftarrow 0$
7: **for** $i = 1$ **to** $|\mathcal{L}_{\text{dp}}|$ **do**
8:     Let $\ell$ be the $i$-th layer in $\mathcal{L}_{\text{dp}}$
9:     **for** $w = 0$ **to** $B$ **do**
10:       **for** each $b_l \in \mathcal{B}$ **do**
11:          $w' \leftarrow w + \Delta w_\ell \cdot b_l$
12:          **if** $w' \leq B$ **then**
13:             $\text{DP}[i][w'] \leftarrow \min(\text{DP}[i][w'], \text{DP}[i-1][w] + \Delta L_\ell(b_l))$
14:          **end if**
15:       **end for**
16:     **end for**
17: **end for**
18: Backtrack from $\arg\min_{w \leq B} \text{DP}[|\mathcal{L}_{\text{dp}}|][w]$ to recover allocation $C_{\text{dp}}$.
19: **return** $C_{\text{dp}}$

---

FLOPs and memory consumption ($\geq 99\%$). Therefore, we keep the embedding section and the final layer at FP without quantization. Within the DiT block, the MLP and adaLN-zero modules occupy most of the memory($\geq 77\%$), while the MLP and attention components dominate the FLOPs($\geq 99\%$). When categorized by computation type, the primary computations occur between weights and activations($\geq 96\%$). In contrast, operations between activations and activations constitute a small proportion but have a significant impact, so we maintain these operations at **8-bit** precision.

C.2   CALCULATE FLOPS OF ROBUQ W1.58A4 MODEL

We employ FLOPs as metrics for evaluating theoretical inference efficiency. For quantization operations, we define the weighted FLOPs as follows:

$$\text{FLOPs}(W = 1.58, A = N) = \frac{1}{2} \cdot \text{FLOPs}(W = N, A = N) = \frac{N}{32} \cdot \text{FLOPs} \tag{37}$$

For the Hadamard transform, since it possesses a fast algorithm with $\mathcal{O}(n^2 \log n)$ complexity (Yates, 1968; Fino & Algazi, 1976) and can be absorbed into the weight matrix within a DiT block—ultimately requiring only four online Hadamard transforms—its theoretical computational cost is

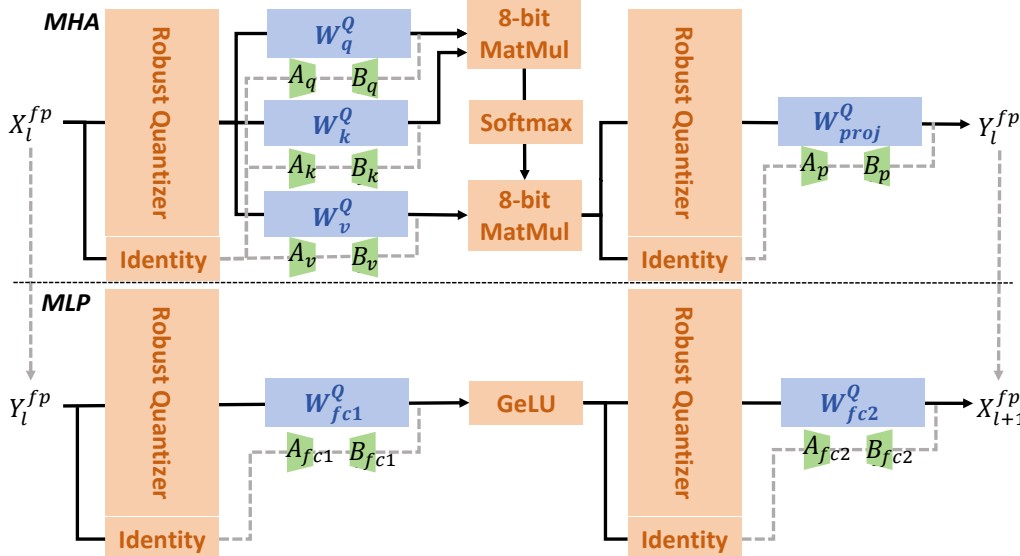

Figure 7: Schematic diagram of actual deployment. For simplicity, we have omitted the AdaLN-zero components to highlight the sections accounting for the majority of FLOPs.

negligible. We provide FLOPs breakdown in RobuQ (w/o AMP) W1.58A4 model as Table 4 shown.

Table 4: Flops breakdown in RobuQ (w/o AMP) W1.58A4 DiT-XL/2 model.

|  | Embedding | Low rank branch | A-A Matrix Multiplication | W-A Matrix Multiplication | Final Layer | Total |
|---|---|---|---|---|---|---|
| bits/bits | 32/32 | 32/32 | 8/8 | 1.58/4 | 32/32 | N/A |
| GFLOPs (G) | 0.0016 | 2.0312 | 1.0133 | 6.9213 | 0.1268 | 10.07 |

Table 5: Deployment on NVIDIA RTX 4090. Iteration Speed is calculated with batchsize=1.

| Method | W/A | FLOPs | Checkpoint Size | Max Memory Allocated | Iteration Speed |
|---|---|---|---|---|---|
| FP | 32/32 | 114.52 | 2.58GB | 3,914MB | 44.09 iter/s |
| RobuQ | 1.58/4 | 10.07 | 0.17GB | 554MB | 155.37 iter/s |
| RobuQ (w/o Hadamard) | 1.58/4 | 10.07 | 0.17GB | 551MB | 168.42 iter/s |

### C.3 INTEGRATION OF THE HADAMARD TRANSFORM DURING INFERENCE

For completeness, we describe how the Hadamard transform is incorporated into the inference pipeline. In our implementation, the transform is applied online only to the activations of *mlp.fc2* and *attn.proj*, while all other occurrences can be absorbed into the weights of preceding layers, ensuring no additional runtime overhead elsewhere. We implement the operation using the **fast-hadamard-transformer** library, where the Hadamard computation is largely memory-bound and typically accounts for only 5%–10% of the total time of the associated matrix operations.

Although additional speedups are achievable—for example, by leveraging the butterfly structure of the Hadamard transform to fuse it with subsequent low-bit operators—such improvements require specialized kernel engineering and are left for future optimization. Given the strong dependency of end-to-end latency on these implementation details and on specific hardware backends, we instead provide a high-level description of the integration strategy without reporting a concrete latency measurement.

### C.4 DEPLOYMENT

Our proposed RobuQ scheme is primarily motivated by the pursuit of enhanced deployment efficiency. To validate its effectiveness, we conducted comparative experiments on DiT-XL/2 under two precision settings: full-precision and a quantized configuration with W1.58A4. In the quantized implementation, we employ a weight-packing strategy that compacts five ternary weights into a Int8, with real-time unpacking to 4-bit weights during inference using SVDQuant's (Li et al., 2025) Nunchaku framework, while additionally introducing an online Hadamard transform following the

approach illustrated in Fig 7.As shown in Table 5, our method achieves a model compression ratio of 15.2×, reduces peak GPU memory usage by 7.1×, and accelerates inference by 3.5× compared to the FP baseline. However, due to hardware limitations and the lack of specialized optimization in current quantization libraries, the acceleration potential has not been fully realized. To further isolate the influence of the Hadamard transform on end-to-end inference latency, we additionally report the empirical results of a variant that removes the Hadamard operation. We believe that as the community continues to refine and develop dedicated low-bit computing frameworks, our approach will demonstrate even greater advantages.

# D  ADDITIONAL ANALYSIS

## D.1  EXPERIMENTS WITH EQUAL BIT-WIDTH (W=A)

We further evaluate our method in an equal bit-width setting, where both weights and activations are quantized to the same precision. For these experiments, we adopted the same configuration as our main experiments, but trained for a shorter duration of 25k steps. As shown in Table 6, our approach consistently surpasses existing baselines across different configurations. In the W4A4 regime, our method significantly improves generative quality over strong competitors such as QueST (Wang et al., 2025a), demonstrating that our quantizer provides tangible benefits even in relatively high-bit settings. More importantly, in the more challenging W3A3 case, our method achieves the best results across all evaluation metrics, confirming its robustness in ultra-low-bit regimes. These results highlight our framework's versatility and SOTA performance under diverse quantization constraints.

Table 6: Performance comparison on ImageNet $256{\times}256$. Timesteps are 50 and cfg is 1.5.

| Method | W/A | IS↑ | FID↓ | sFID↓ | Precision↑ |
|---|---|---|---|---|---|
| FP | 32/32 | 239.50 | 6.62 | 21.10 | 0.7849 |
| QueST | | 4.87 | 215.06 | 72.15 | 0.0529 |
| Baseline | 4/4 | 162.27 | 12.47 | 25.16 | 0.6825 |
| **RobuQ (w/o AMP)** | | **184.51** | **9.91** | **22.96** | **0.7183** |
| Baseline | 3/3 | 78.78 | 30.33 | 34.24 | 0.5604 |
| **RobuQ (w/o AMP)** | | **143.82** | **14.67** | **26.8** | **0.6844** |

## D.2  PTQ RESULTS

For completeness, we report additional PTQ results under the W4A4 configuration, as shown in Table 7, comparing our method against several representative PTQ baselines, including Q-Diffusion (Li et al.), Quarot (Ashkboos et al., 2024), and SVD-Quant (Li et al., 2025). All models are evaluated using cfg = 1.5, 50 sampling steps, and 10k evaluation images. These methods exhibit diverse behaviors across different evaluation metrics. Q-Diffusion suffers from significant performance degradation in both fidelity and perceptual quality metrics, while Quarot and SVD-Quant achieve reasonably stronger results but still fall short of our method. In contrast, our approach maintains competitive perceptual quality while substantially outperforming all baselines across IS, FID, and sFID, demonstrating the robustness of our activation quantization strategy even under equal bit-width constraints. For SVDQuant, since it is a per-group scheme, we implemented its per-channel version for fair comparison.

Table 7: Comparison of PTQ baselines under the W4A4 setting.

| Method | W/A | IS ↑ | FID ↓ | sFID ↓ | Precision ↑ |
|---|---|---|---|---|---|
| FP | 32/32 | 239.50 | 6.62 | 21.10 | 0.7849 |
| Q-Diffusion | | 1.37 | 304.12 | 175.88 | 0.0048 |
| Quarot | 4/4 | 53.12 | 53.31 | 56.74 | 0.4134 |
| SVD-Quant | | 55.23 | 50.47 | 44.26 | 0.4313 |
| **Ours** | | **155.96** | **14.12** | **26.32** | **0.6779** |

## D.3  ABLATION STUDY ON SVD RANK

To further understand the impact of the low-rank decomposition used in our weight processing pipeline, we conduct an ablation study by varying the SVD rank under the W1.58A4 configuration. All models are trained for 200k steps using 10k samples (cfg = 1.5, step size = 50), and evaluated under the same protocol as our main experiments. The results are reported in Table 8.

We observe that increasing the rank consistently improves generative quality, especially in IS and FID. However, the marginal gain becomes smaller when moving from rank 16 to rank 32, while

Table 8: Ablation study on different SVD ranks for W1.58A4 quantization.

| Rank | IS ↑ | FID ↓ | sFID ↓ | Precision ↑ |
|---|---|---|---|---|
| 8 | 87.43 | 23.21 | 29.96 | 0.6327 |
| 16 | 91.28 | 22.25 | 29.24 | 0.6328 |
| 32 | 92.11 | 21.71 | 28.89 | 0.6324 |

the computational overhead continues to increase. Based on this trade-off between accuracy and efficiency, we adopt rank = 16 in all main experiments.

## D.4 COMPARISON OF UNIFORM AND NON-UNIFORM QUANTIZERS ON T2I MODELS

To further examine whether the advantages of our Gaussian-optimized non-uniform quantizer persist when applied to larger and more advanced text-to-image (T2I) models, we conduct an additional experiment on a recent state-of-the-art open-source model, FLUX (Labs, 2024). Specifically, we evaluate the Flux-Schnell variant using 4 sampling steps on 5k images from the MJHQ dataset (Huang et al., 2024). All settings remain consistent across quantizers to ensure a fair comparison. The quantitative results are reported in Table 9.

Table 9: Comparison between uniform and non-uniform quantizers on FLUX-Schnell.

| Quantizer | FID ↓ | Image Reward ↑ | CLIPIQA ↑ | CLIPScore ↑ | PSNR ↑ |
|---|---|---|---|---|---|
| FP | 18.40 | 0.9323 | 0.9399 | 26.54 | N/A |
| Uniform | 18.57 | 0.8617 | 0.9266 | 26.34 | 17.21 |
| Non-Uniform | **18.49** | **0.8698** | **0.9296** | **26.41** | **17.34** |

Across all metrics, the non-uniform quantizer demonstrates a mild but consistent improvement over the uniform quantizer, aligning with our theoretical motivation based on Gaussianity after the Hadamard transform. These gains, however, remain relatively modest in magnitude.

In practice, the choice of quantizer must also consider factors beyond accuracy alone. Compared to its non-uniform counterpart, the uniform quantizer exhibits stronger robustness across heterogeneous hardware backends, simpler implementation, and lower computational overhead—particularly for high-throughput inference workloads. Taking these engineering considerations into account, we ultimately select the **Uniform Quantizer** as the preferred option for real-world deployment, despite the slight empirical advantage of non-uniform quantization on larger models such as FLUX.

## D.5 MIXED-PRECISION ANALYSIS

**Setup.** With *adaLN* fixed to 4-bit, we examine activation bit allocation only on learnable layers. Fig. 8 (left) shows per-block heatmaps for the four ops (*attn.qkv*, *attn.proj*, *mlp.fc1*, *mlp.fc2*) under two activation budgets, W1.58A2 and W1.58A3. Fig. 8 (right) summarizes the mean per-op allocation via pie charts, while Fig. 9 (top) and Fig. 9 (bottom) plot, respectively, the per-block average activation bits and the normalized per-block activation share.

**Findings.**

- **Attention consumes the budget first.** From the heatmaps (Fig. 8 (left)), attention paths (*attn.qkv*, *attn.proj*) retain higher bitwidths in mid and late blocks under W1.58A2. When moving from A2 to A3, the bit-width allocation becomes more balanced across all ops, with extra capacity used to maintain higher precision, although attention still dominates.

- ***attn.proj* is the largest sink under tight budgets.** The pies in Fig. 8 (right) show that at A2, *attn.proj* receives the largest share (32.0%), with *attn.qkv* and *mlp.fc1* close behind. At A3, the distribution becomes more even, but attention still takes the largest share.

- **Depth matters: later blocks require more bits.** The per-block curves (Fig. 9 (top)) increase with depth for both budgets, and the share curves (Fig. 9 (bottom)) peak in the middle-to-late stages, indicating that deeper layers need more precision for stability.

- **A3 mainly lifts the floor.** Upgrading from A2 to A3 shifts the entire per-block curve upward (Fig. 9 (top)), reducing the number of low-precision stretches in both attention and MLP. This suggests a robustness effect: more bits can smooth the activation statistics.

**Practical rules-of-thumb.** These observations can guide activation mixed precision policies.

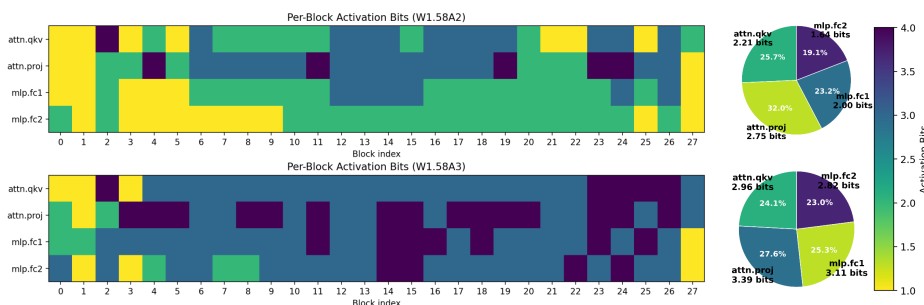

Figure 8: Visualization of Activation Bit-Width Distribution

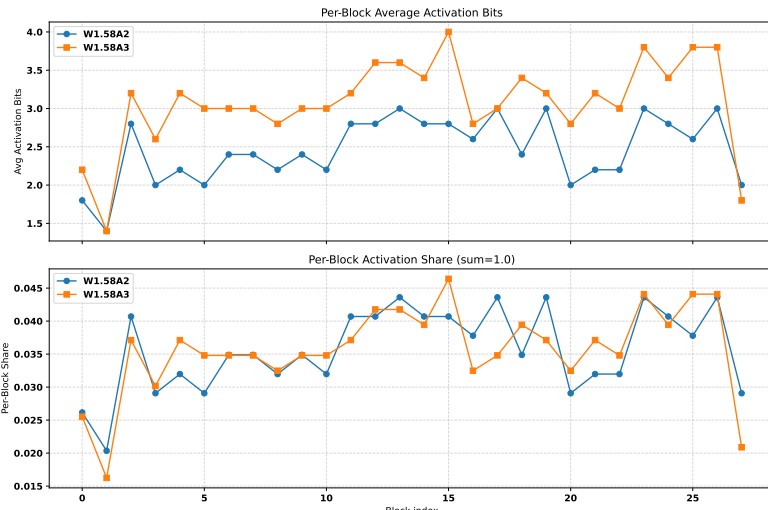

Figure 9: Per-Block Activation Statistics. Top: average activation bits per block; Bottom: normalized per-block share (sums to 1).

- **Prioritize attention first, projection before QKV under tight budgets.** If only a small headroom is available, raise *attn.proj* and then *attn.qkv*.

- **Bias budget to mid/late blocks.** Allocate extra bits to the second half of the network, where attention–attention interactions accumulate and feature distributions widen.

### D.6 VISUALIZATION RESULTS COMPARISONS

We present additional visualization results from our DiT-XL/2 model at a $256 \times 256$ resolution, using 250 sampling steps (Ho et al., 2020) and a cfg of $= 4.0$ (Ho & Salimans, 2022). Figure 10 compares three activation precision configurations (A4, A3, and A2) across different quantization methods, displaying W1.58 DiT-XL/2 samples for ImageNet (Russakovsky et al., 2015). **More visualizations can be found in the supplementary materials.**

## E STATEMENT ON LARGE LANGUAGE MODEL USAGE

In preparing this manuscript, the authors used GPT-5 solely for language editing to improve readability and clarity. Typical interactions included suggestions on grammar, syntax, style, and concision; harmonization of terminology and notation across sections; and minor rephrasings to enhance narrative flow. All suggested edits were reviewed by the authors line by line, and acceptance required human verification for factual accuracy and technical fidelity.

The model was not used for research ideation, experiment design, data processing or analysis, figure or table generation, or drawing scientific conclusions. No code was authored, debugged, or executed by the model, and no images were created or altered using generative tools. All methods and results reported here were developed, implemented, and validated independently by the authors.

Throughout the writing process, the authors maintained control over scientific content and ensured that any language edits did not alter the technical meaning. No confidential or proprietary information beyond the manuscript text itself was provided to the model. The authors remain fully responsible for the integrity and correctness of the paper and for any errors or omissions therein.

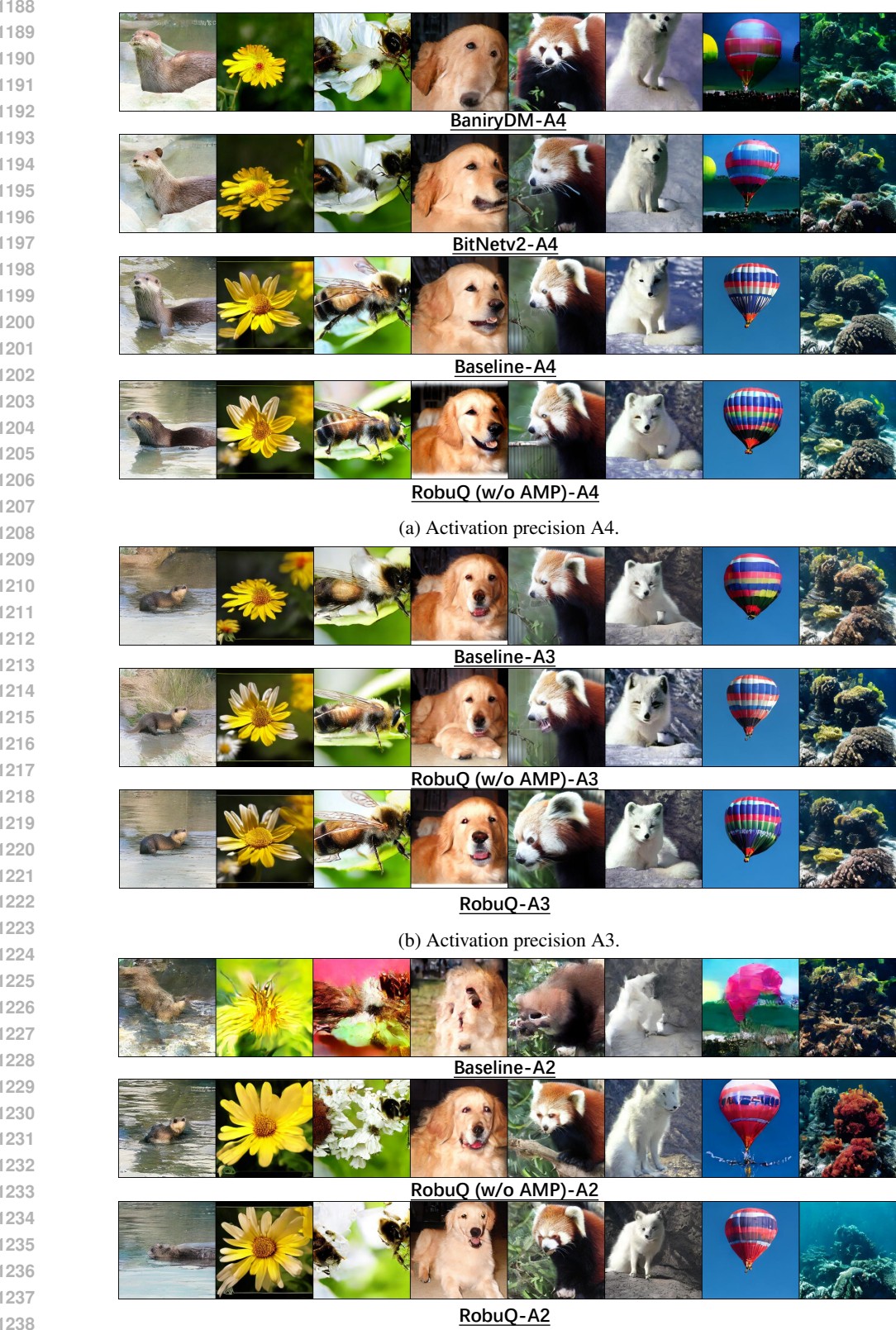

(a) Activation precision A4.

(b) Activation precision A3.

(c) Activation precision A2.

Figure 10: W1.58 DiT-XL/2 samples at $256 \times 256$. Labels = [360, 985, 309, 207, 387, 279, 417, 973]. Cfg = 4.0, sampling steps = 250.

