# RobuQ: Pushing DiTs to W1.58A2 via Robust Activation Quantization

## 1 Additional Visualization Results

We provide additional qualitative results from DiT-XL/2 (Peebles & Xie, 2023) at 256×256 resolution. Figures 1a and 1b show samples from W1.58A2 and W1.58A4, and Figures 2–11 compare RobuQ (w/o AMP)-A4, RobuQ-A3, and RobuQ-A2, highlighting consistent quality and robustness across activation precisions. All samples are generated with 250 steps Ho et al. (2020) and cfg (Ho & Salimans, 2022) of 4.0 on ImageNet-1K (Russakovsky et al., 2015) dataset.

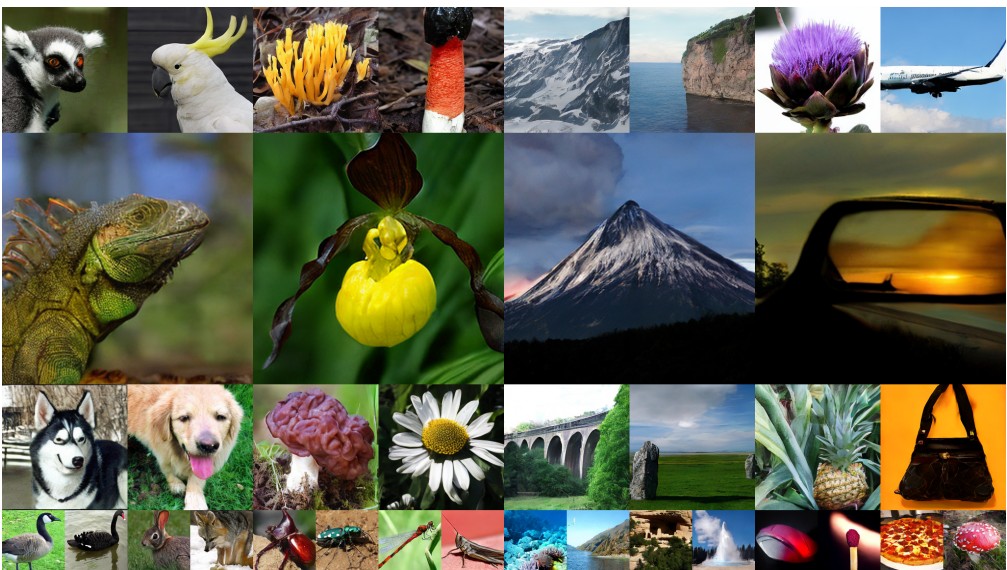

(a) RobuQ-A2

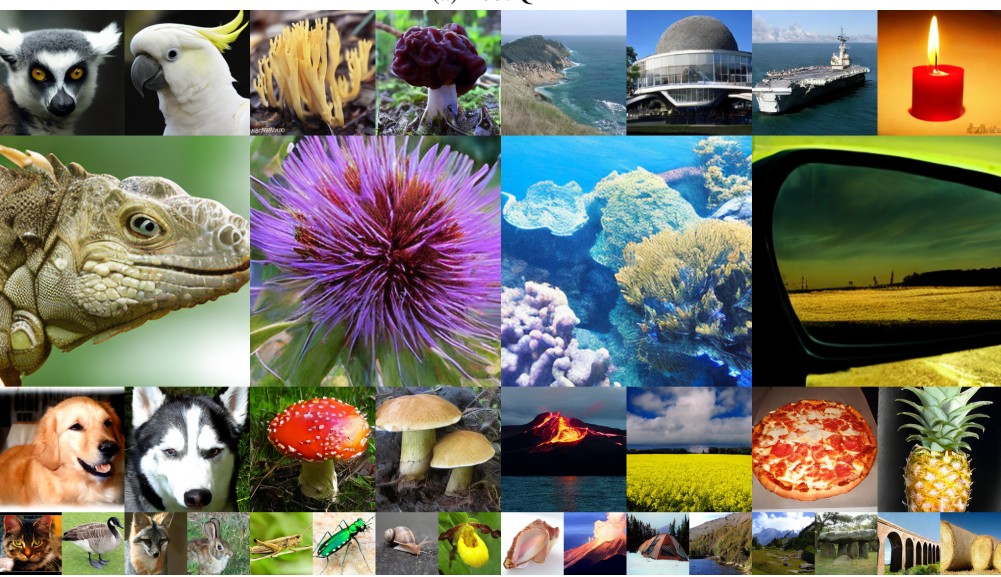

(b) RobuQ (w/o AMP)-A4

Figure 1: **RobuQ enables DiTs to generate competitive results at ultra-low bit settings.** We show 256×256 samples from W1.58A2 and W1.58A4 quantized models trained on ImageNet-1K.

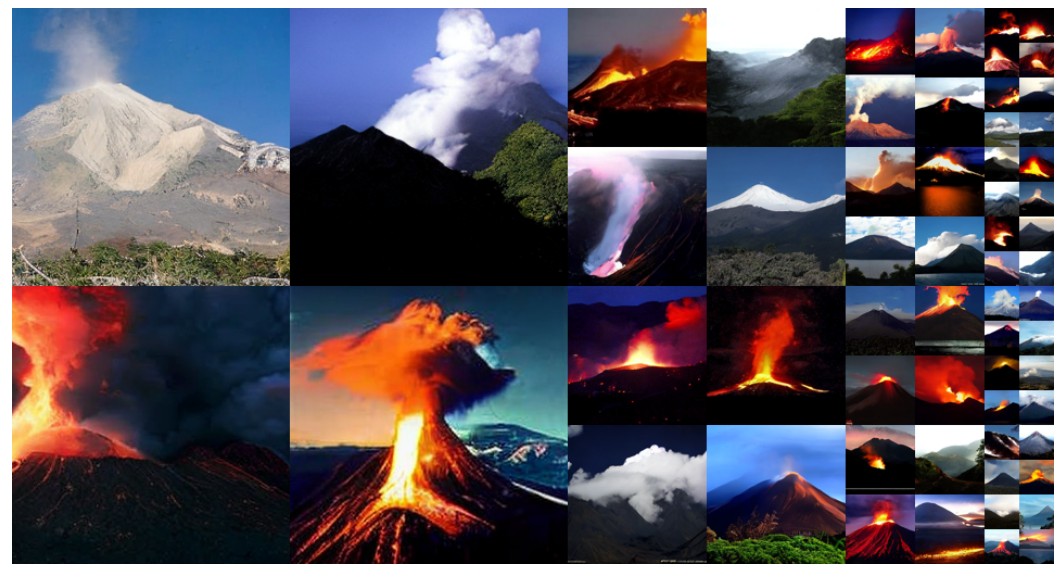

(a) RobuQ (w/o AMP)-A4

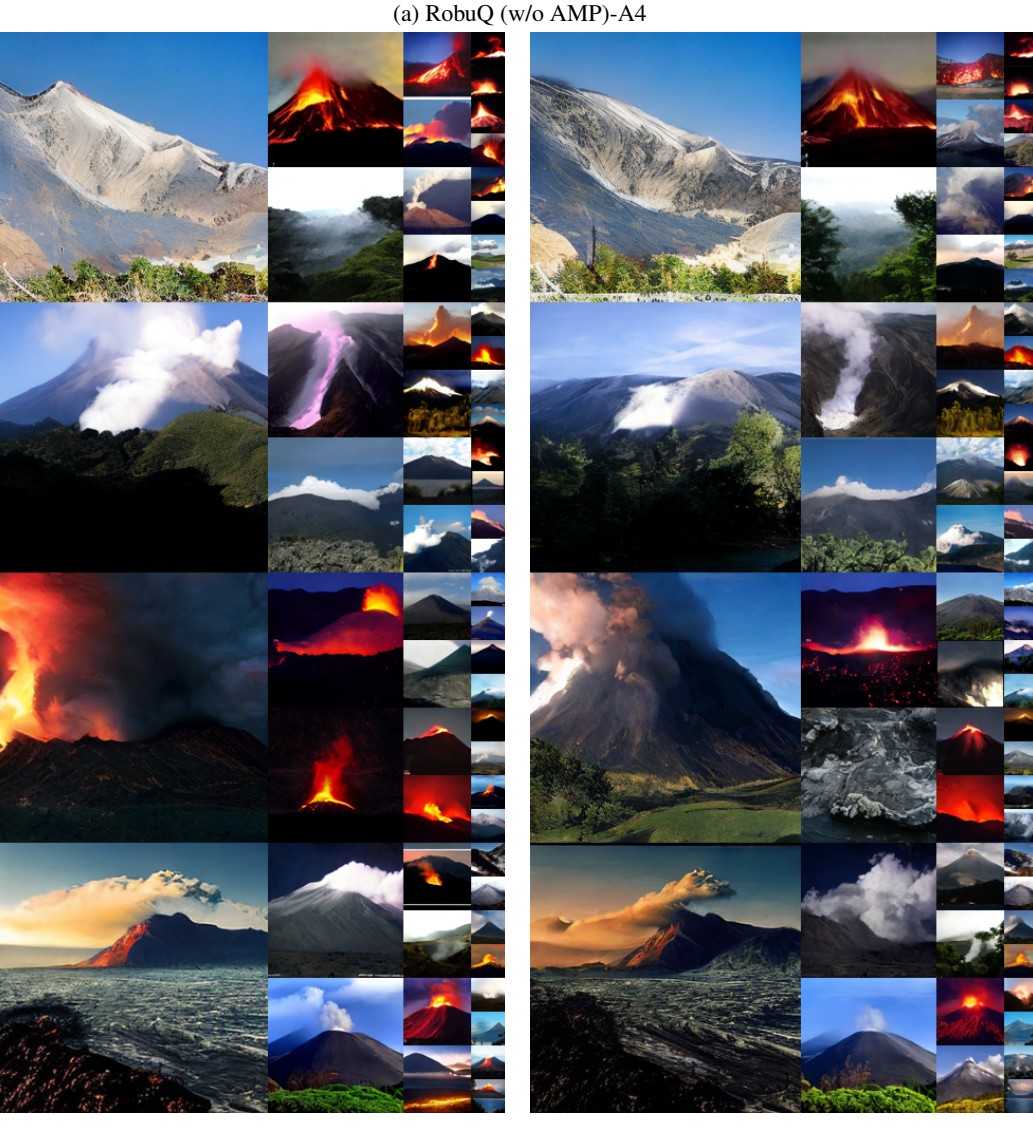

(b) RobuQ-A3

(c) RobuQ-A2

Figure 2: Uncurated DiT-XL/2 256×256 samples, cfg = 4.0, class label = 980

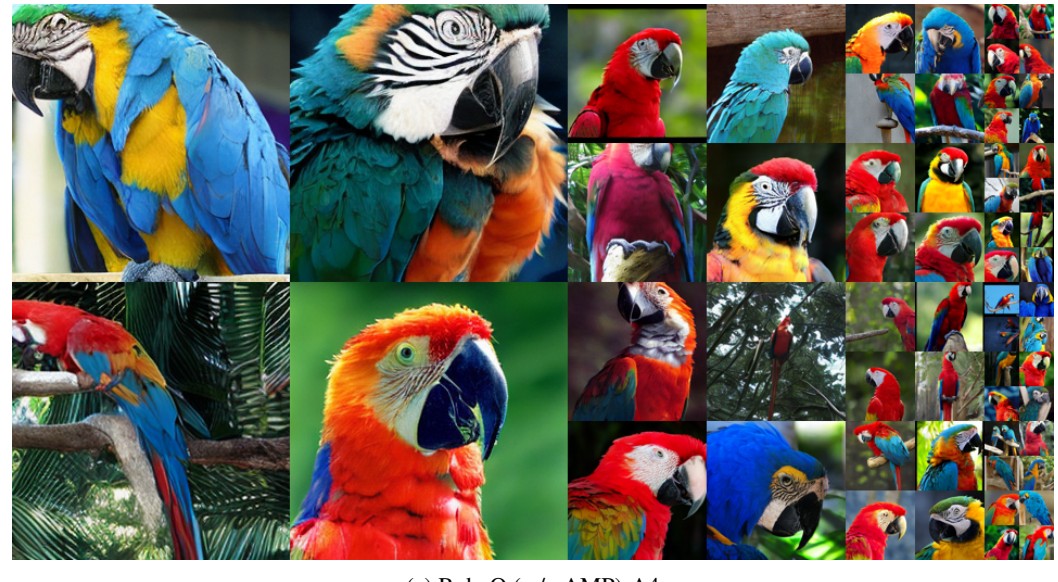

(a) RobuQ (w/o AMP)-A4

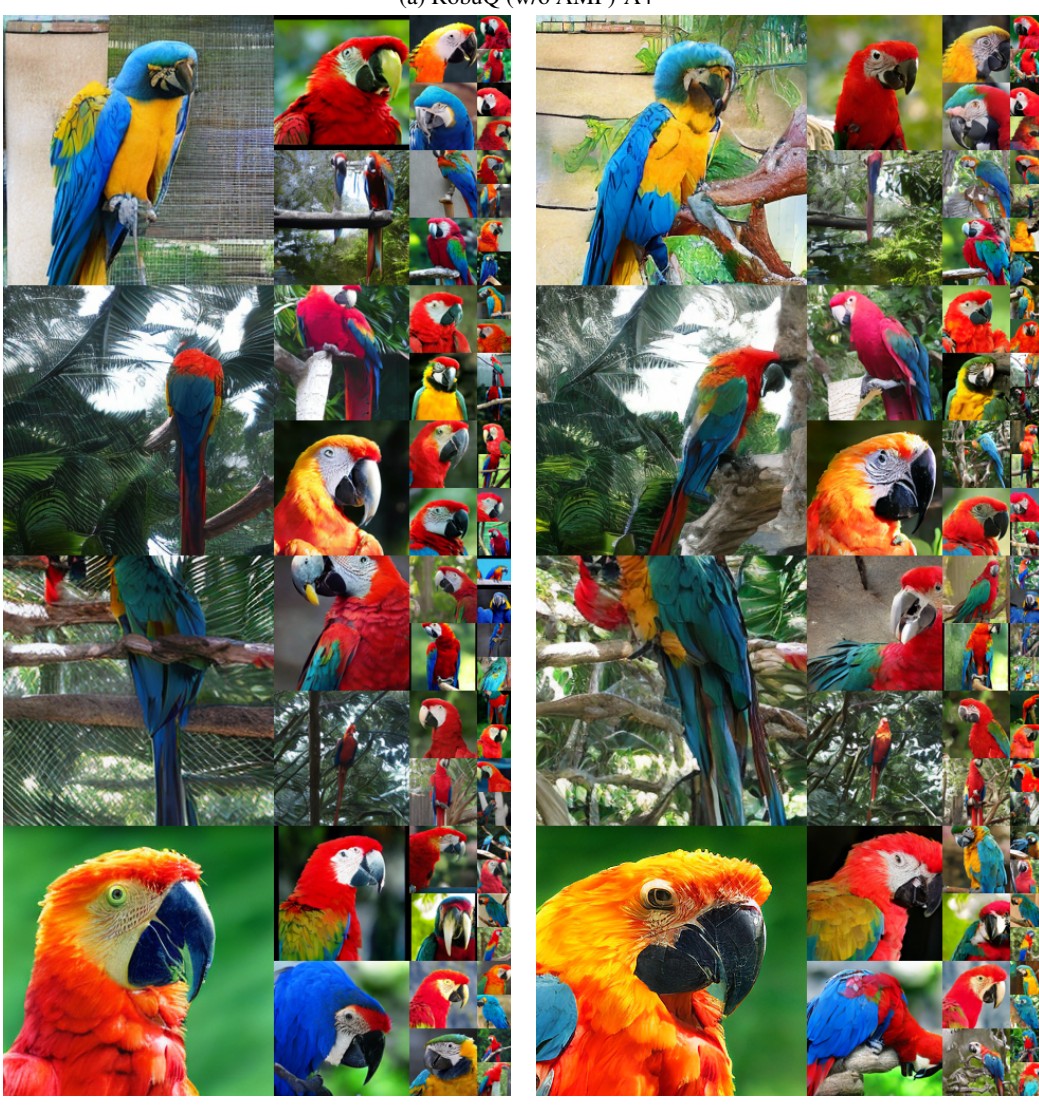

(b) RobuQ-A3                         (c) RobuQ-A2

Figure 3: Uncurated DiT-XL/2 256×256 samples, cfg = 4.0, class label = 88

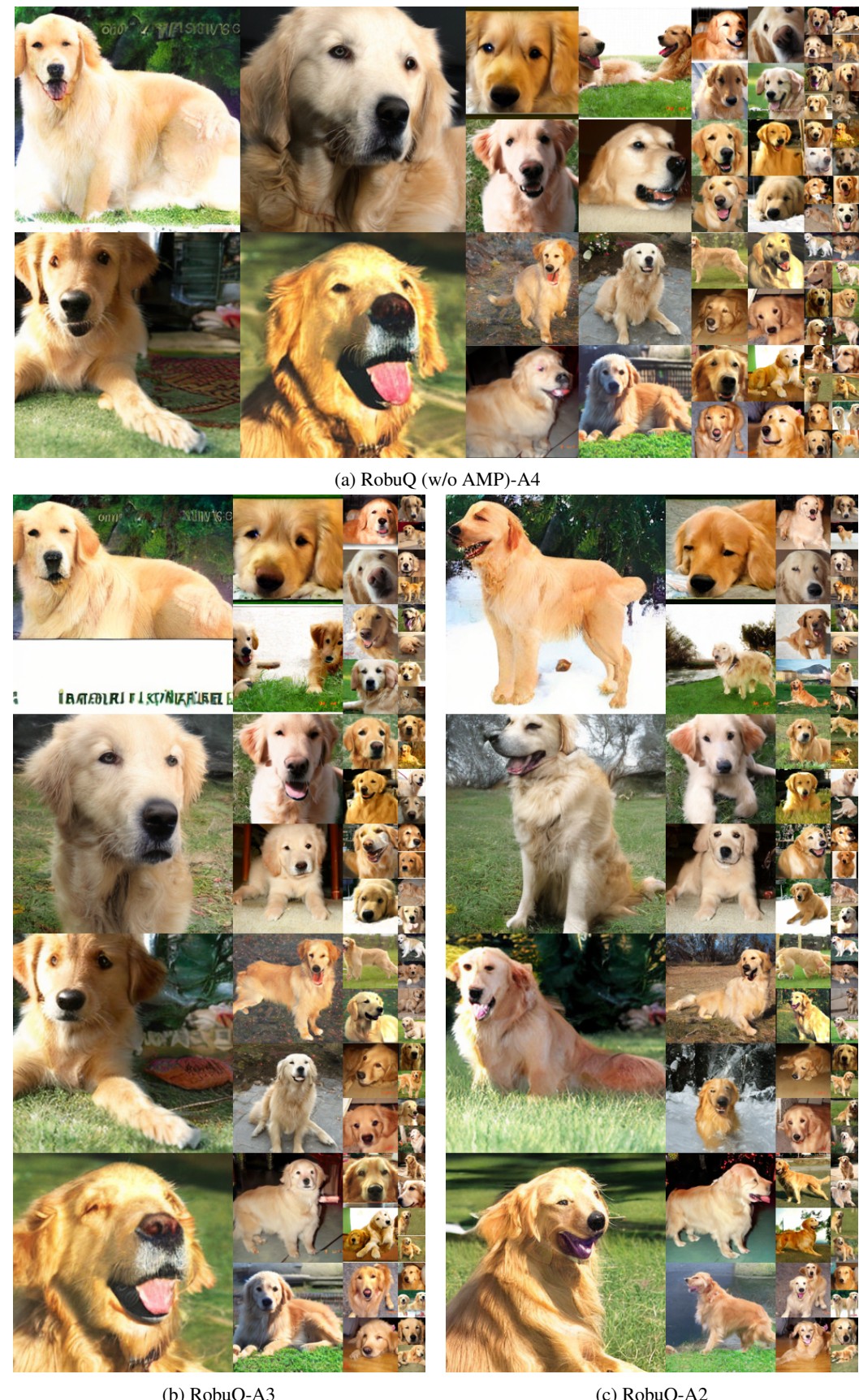

(a) RobuQ (w/o AMP)-A4

(b) RobuQ-A3        (c) RobuQ-A2

Figure 4: Uncurated DiT-XL/2 $256{\times}256$ samples, cfg = 4.0, class label = 207

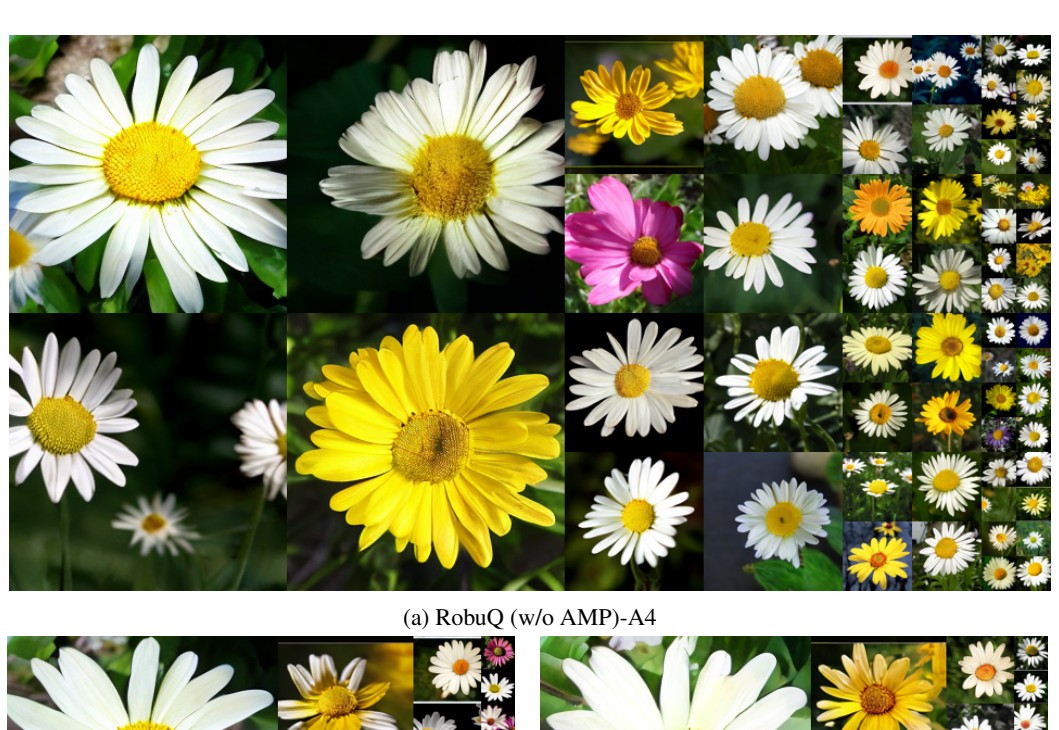

(a) RobuQ (w/o AMP)-A4

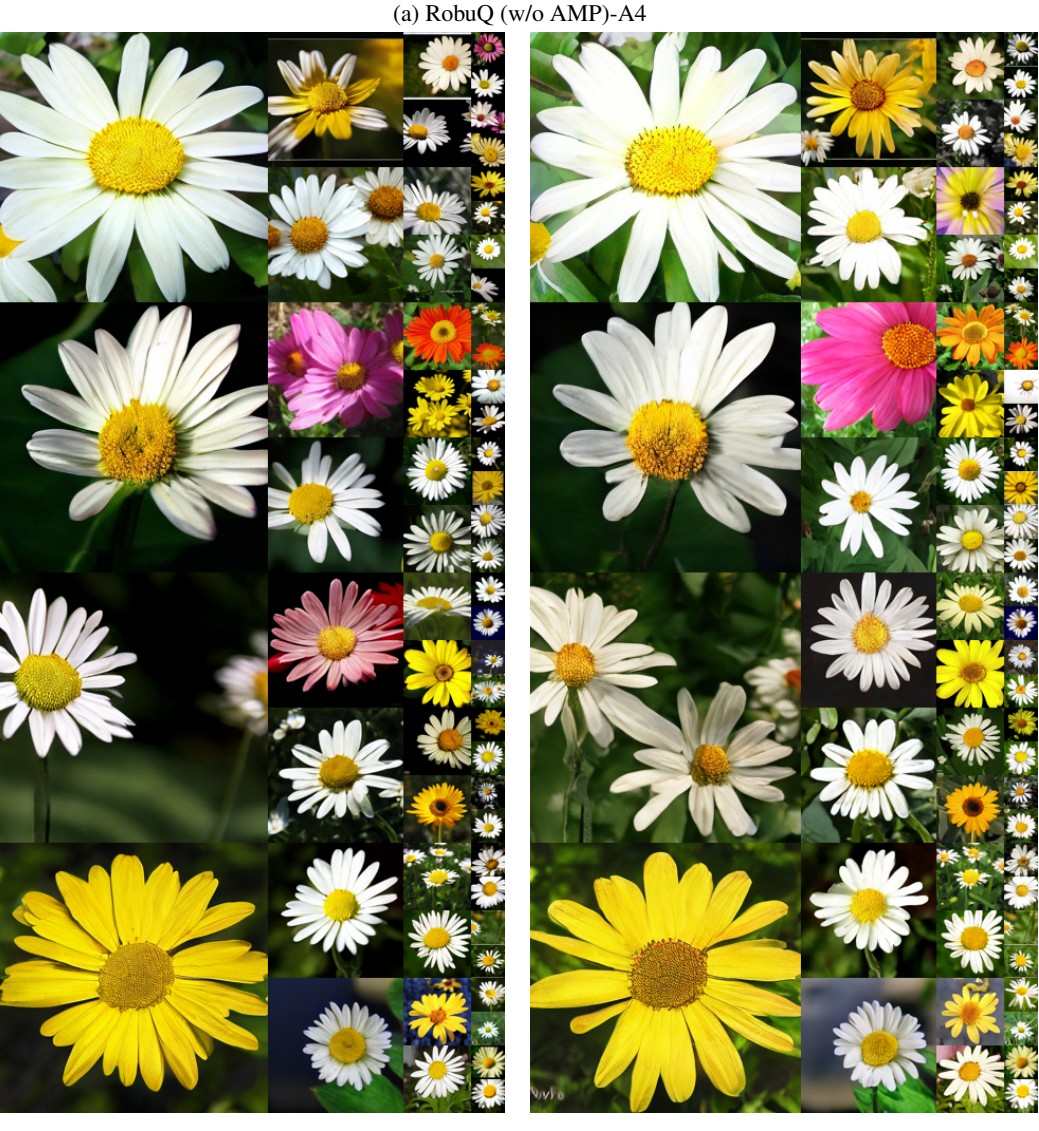

(b) RobuQ-A3          (c) RobuQ-A2

Figure 5: Uncurated DiT-XL/2 $256 \times 256$ samples, cfg = 4.0, class label = 985

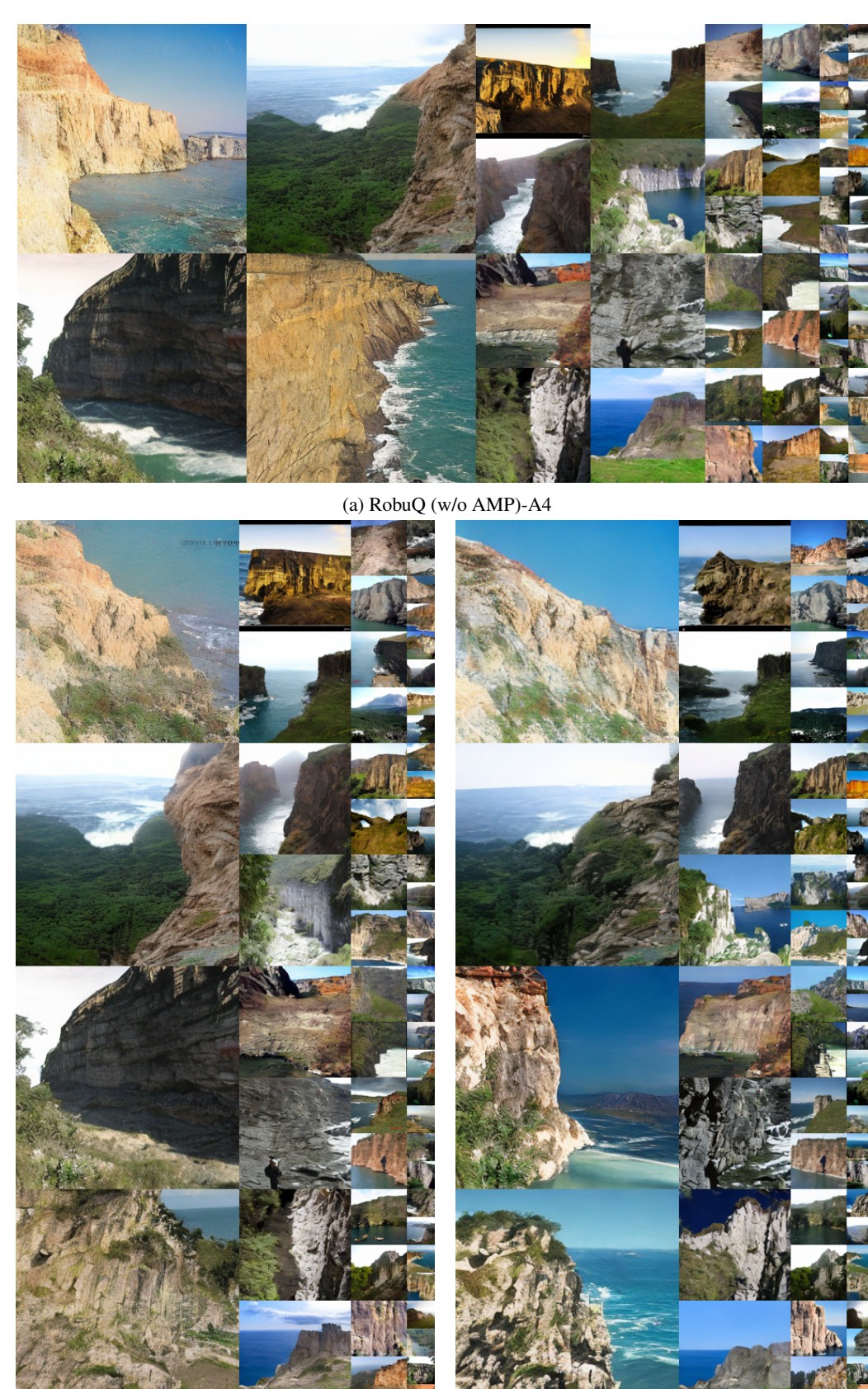

(a) RobuQ (w/o AMP)-A4

(b) RobuQ-A3

(c) RobuQ-A2

Figure 6: Uncurated DiT-XL/2 256×256 samples, cfg = 4.0, class label = 972

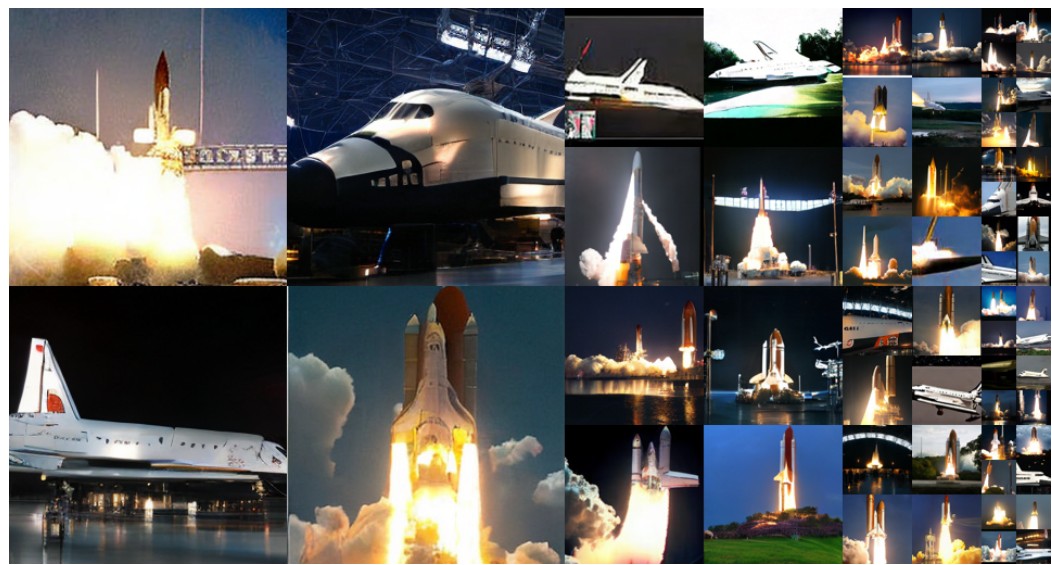

(a) RobuQ (w/o AMP)-A4

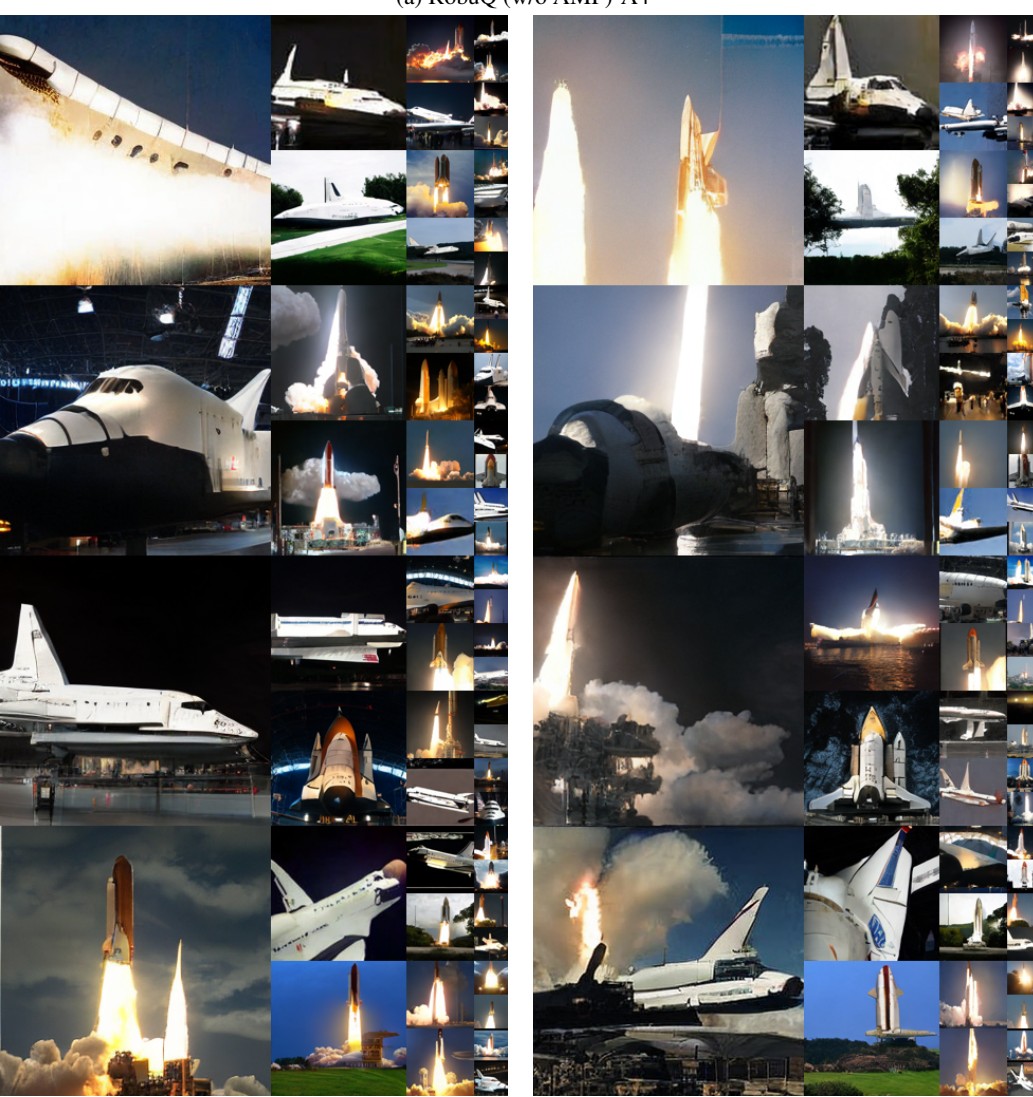

(b) RobuQ-A3          (c) RobuQ-A2

Figure 7: Uncurated DiT-XL/2 256×256 samples, cfg = 4.0, class label = 812

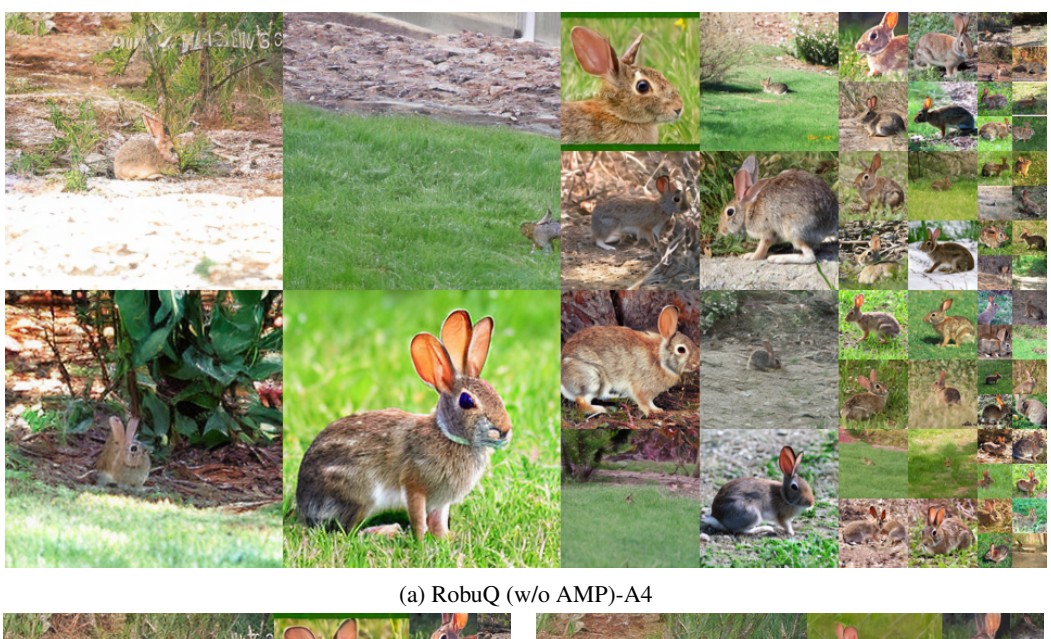

(a) RobuQ (w/o AMP)-A4

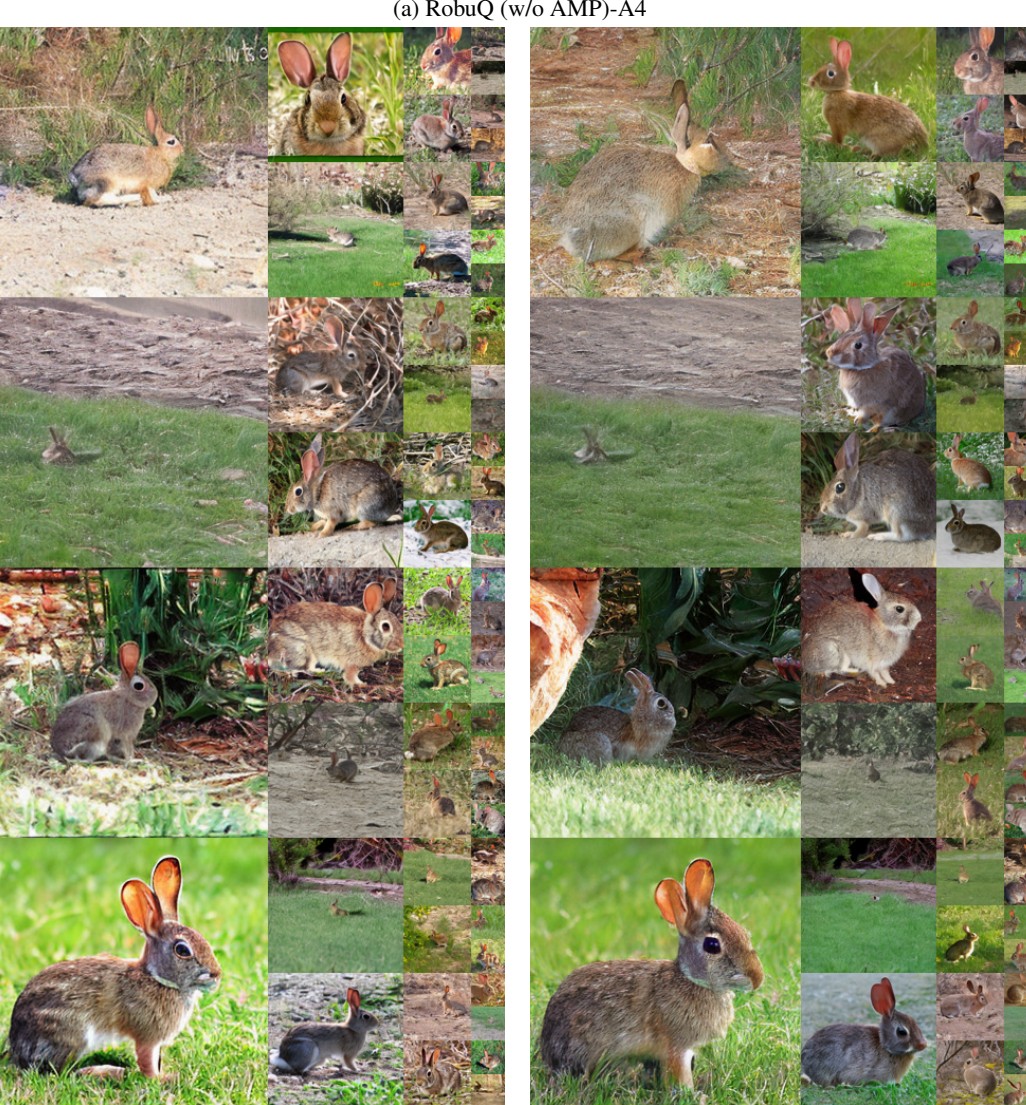

(b) RobuQ-A3                                    (c) RobuQ-A2

Figure 8: Uncurated DiT-XL/2 256×256 samples, cfg = 4.0, class label = 330

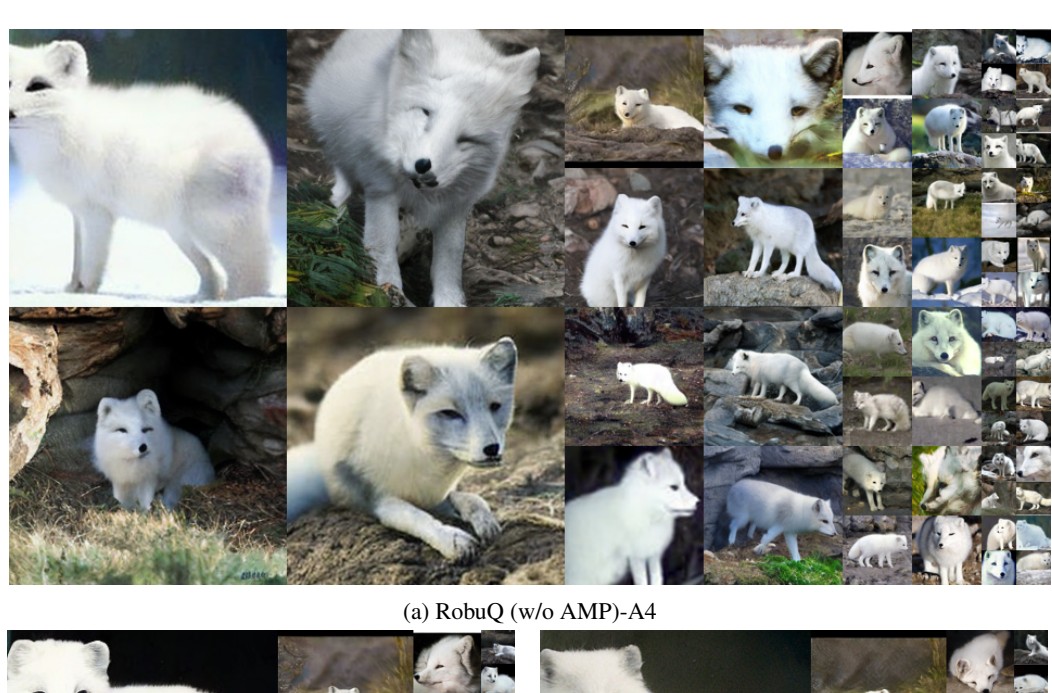

(a) RobuQ (w/o AMP)-A4

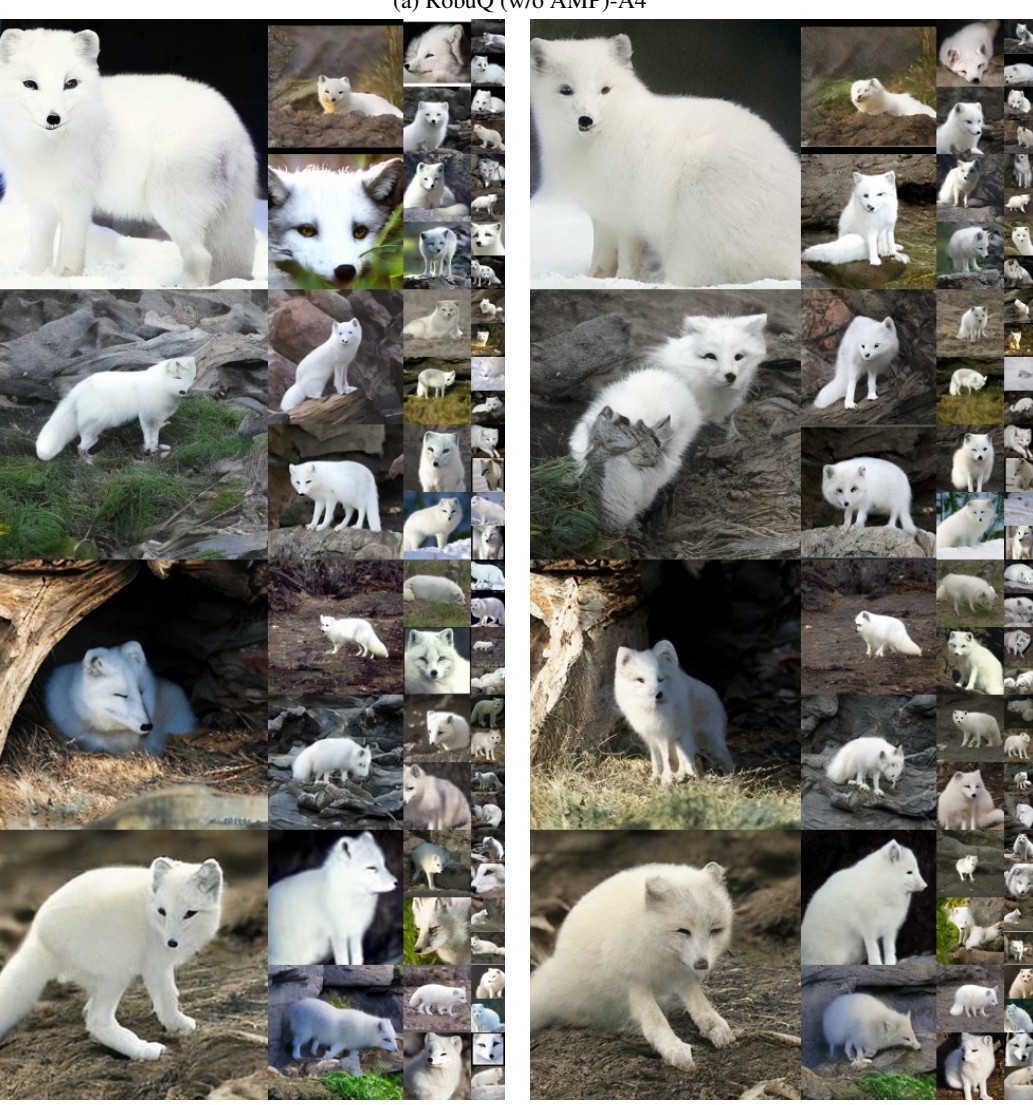

(b) RobuQ-A3

(c) RobuQ-A2

Figure 9: Uncurated DiT-XL/2 256×256 samples, cfg = 4.0, class label = 279

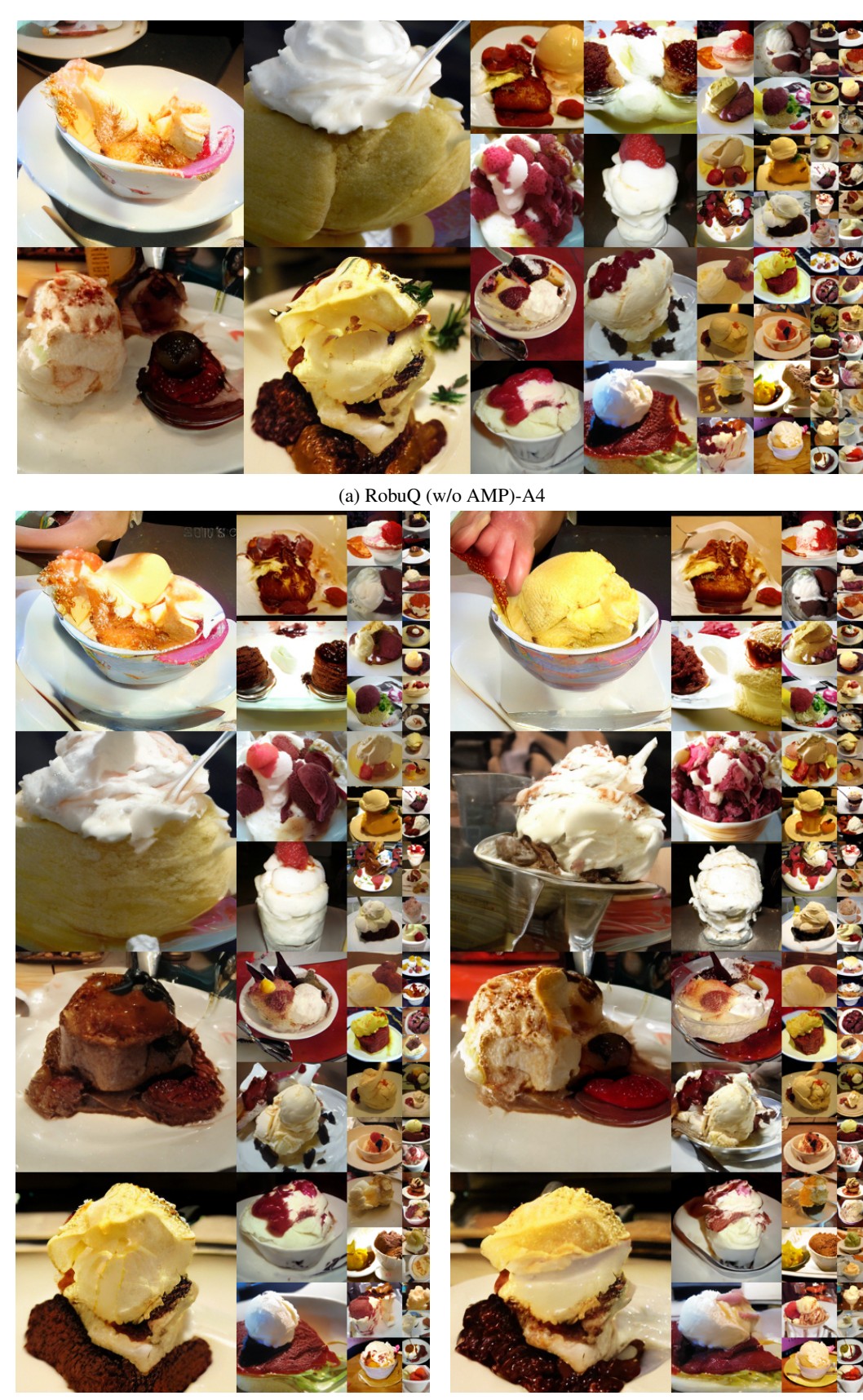

(a) RobuQ (w/o AMP)-A4

(b) RobuQ-A3

(c) RobuQ-A2

Figure 10: Uncurated DiT-XL/2 256×256 samples, cfg = 4.0, class label = 928

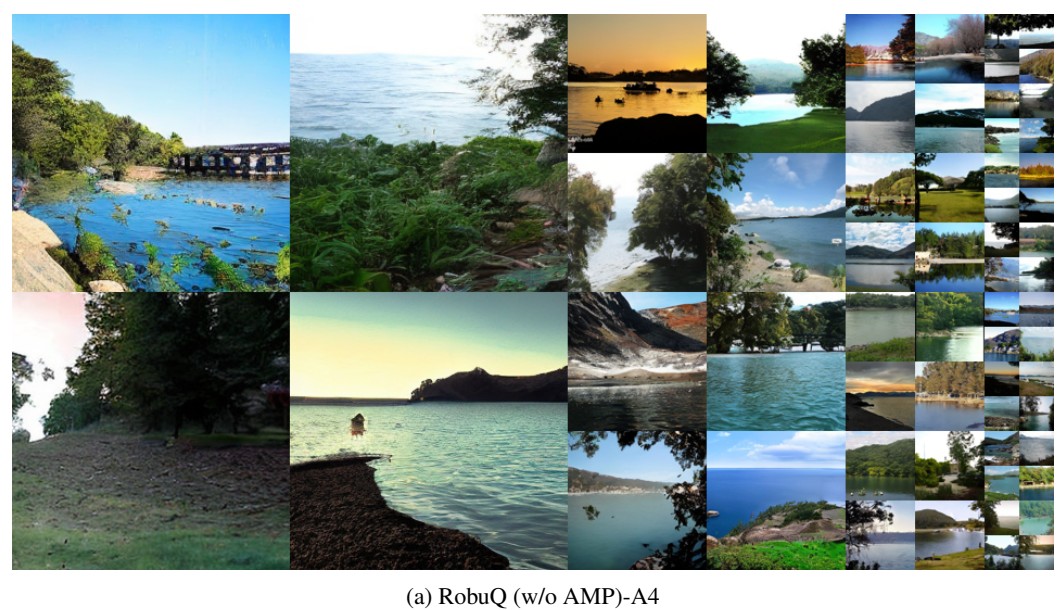

(a) RobuQ (w/o AMP)-A4

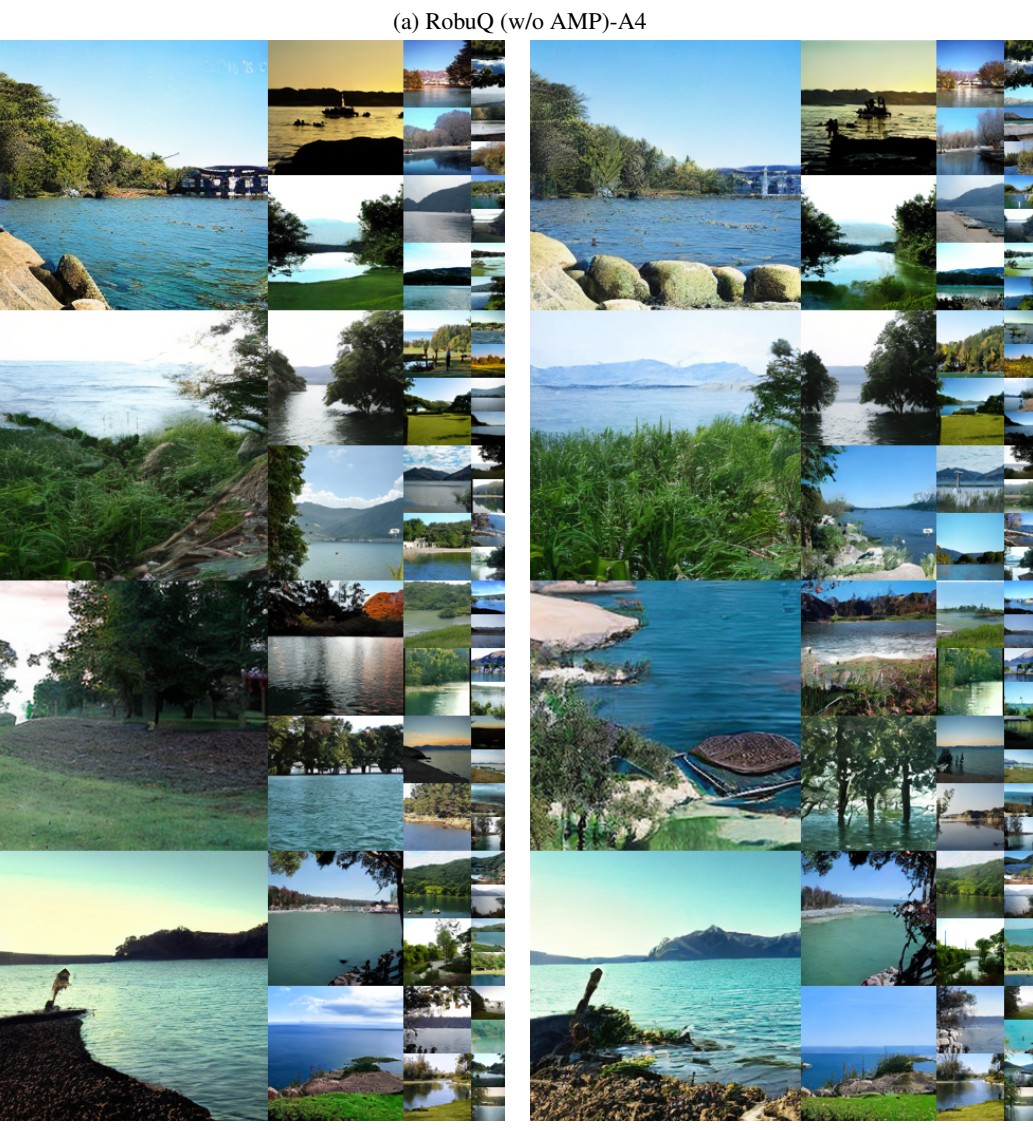

(b) RobuQ-A3

(c) RobuQ-A2

Figure 11: Uncurated DiT-XL/2 256×256 samples, cfg = 4.0, class label = 975