# OpenReview forum: "RobuQ: Pushing DiTs to W1.58A2 via Robust Activation Quantization"
_ICLR.cc/2026/Conference — ICLR 2026 Conference Desk Rejected Submission_

### Official Review · Reviewer_fB1t · 2025-10-29

**Soundness:** 3
**Presentation:** 3
**Contribution:** 3
**Rating:** 8
**Confidence:** 4

**Summary:**

This paper investigates Quantization-Aware Training (QAT) for Diffusion Transformers (DiTs) and proposes a systematic framework named RobuQ, which enables ultra-low-bit quantization (down to W1.58A2) while preserving generative quality.  The authors identify activation quantization as the key bottleneck in DiT compression and introduce three main components:

1.  A baseline established by integrating SVD-initialized low-rank compensation with Hadamard transformation.

2.  A RobustQuantizer, which theoretically transforms per-token activation values into a normal distribution via Hadamard transformation, enabling distribution-agnostic quantization.

3.  An Activation-only Mixed-Precision Network (AMPN), which optimizes activation bit allocation under a target quantization level using dynamic programming.

Experiments on ImageNet-1K and FFHQ demonstrate that RobuQ achieves significant efficiency improvements over existing QAT and PTQ baselines while maintaining competitive image generation quality.

**Strengths:**

1. The paper introduces the Hadamard transform and provides a rigorous proof that this transformation can approximately normalize the channel-wise activation distributions, effectively mitigating quantization errors caused by irregular activation distributions. This insight is leveraged to achieve stable W1.58A2 activation precision in DiTs.

2. The proposed AMPN (Activation-only Mixed-Precision Network) strategy dynamically allocates bit-widths across activation layers, offering valuable insights for model compression and efficient deployment.

3. The manuscript is clearly written and enriched with informative figures and tables. The appendix includes detailed mathematical derivations and comprehensive experimental results, enhancing the overall rigor and reproducibility of the work.

**Weaknesses:**

The paper draws heavily on recent quantization techniques such as BitNetV2, SVD-Quant and BinaryDM. While the integration of these methods yields strong results, the overall contribution would be more compelling if it included additional novel components that are clearly attributable to the authors.

**Questions:**

Beyond the integration of existing techniques, which specific components of RobuQ constitute genuine technical novelty?

---

> ### Author Response · Authors · 2025-11-22
> **Response to Reviewer fB1t(denoted as R4)**
>
> We thank the reviewer for the constructive feedback and for acknowledging the key contributions of our study. We provide our responses to the reviewer’s concerns below.
>
> `Q4-1` The paper draws heavily on recent quantization techniques such as BitNetV2, SVD-Quant and BinaryDM. While the integration of these methods yields strong results, the overall contribution would be more compelling if it included additional novel components that are clearly attributable to the authors. Beyond the integration of existing techniques, which specific components of RobuQ constitute genuine technical novelty?
>
> `A4-1` We appreciate the reviewer’s concern regarding the novelty of our work. We would like to clarify that, although RobuQ builds on several established quantization principles, it also incorporates a number of targeted and technically substantive components developed specifically for our setting. These elements go beyond a simple combination of existing techniques and reflect design choices that address challenges unique to Diffusion Transformers.
>
> The core technical novelties of RobuQ are:
>
> - **Identification of the Activation Bottleneck in DiTs:**
>   Our work is the first to systematically identify and report **activation quantization** as the primary, unique barrier for pushing DiTs into the sub-4-bit regimes, distinguishing the challenge from U-Nets or autoregressive models. This novel problem formulation directed the subsequent development of RobuQ's specialized components.
>
> - **Novel Theoretical Foundation for Activation Quantization:**
>   We are the first to theoretically establish that the **Hadamard transform** can convert arbitrary, highly complex *per-token distributions* in DiT activations into approximately **normal distributions**. This insight provides the principled mechanism for designing our **RobustQuantizer**, allowing for stabilized quantization at extremely low bit-widths where prior methods fail. This theoretical analysis and application of per-token normality are entirely novel and are not present in BitNetV2 or prior DiT quantization literature.
>
> - **First Activation-only Mixed-Precision Network (AMPN) for DiTs:**
>   We introduce AMPN (Activation-only Mixed-Precision Network), which is the first method to address the activation bottleneck by allocating heterogeneous activation bit-widths per layer under a ternary-weight backbone. Existing DiT quantization methods use fixed precision or only focus on weight-based mixed precision. The design and implementation of this activation-specific scheduling mechanism to eliminate information bottlenecks in DiTs is a unique methodological contribution.
>
> - **First to Achieve A2 Performance:**
>   Empirically, RobuQ is the first framework to achieve stable and competitive image generation on large datasets like ImageNet-1K with activations quantized to an average of 2 bits (A2). This state-of-the-art result at an extreme regime confirms the efficacy and necessity of our novel components (RobustQuantizer and AMPN).
>
> In summary, the contributions of RobuQ stem not only from its end-to-end performance, but also from the **theoretical formulation** behind the RobustQuantizer and the **methodological design** of the AMPN framework. These components were developed specifically to address the activation-quantization challenges in DiTs and represent focused contributions from our works.
>
> We sincerely appreciate your thoughtful and professional comments, which significantly contributed to improving the clarity and completeness of our work. Should you have any further questions, we would be glad to provide additional explanations.

---

> > ### Comment · Reviewer_fB1t · 2025-11-25
> > **Reviewer's Response**
> >
> > ### Thank you for the detailed response. ###
> >  After careful review, we acknowledge several highlights in the response, but also believe there is room for further discussion regarding the justification of the method's novelty and the detailed evaluation of the empirical breakthroughs.
> >
> > ---
> >
> > ##  Strengths Acknowledged ##
> > ### We fully acknowledge the authors' reply ###
> > 1.  The reply clearly states that its core theoretical contribution lies in being the first to prove the normalizing effect of the Hadamard transform on the activation distribution of DiT, rather than merely applying it. This elaboration elevates the work from an engineering technique to a principled innovation, fully meeting the standard of "true technological innovation".
> > 2.  Positioning the systematic identification of "activation quantization as the bottleneck for ultra-low-bit DiT" as the primary contribution is a highly persuasive argument. It demonstrates the foresight of the research, showing that the work stems from an insight into deep challenges in the field, rather than being a simple technical stacking.
> >
> >
> >
> > ---
> >
> > ## Areas for Further Discussion ##
> > ### While acknowledging the above contributions, the authors' innovative claims could be further strengthened with additional information. ###
> > * We note that the paper already points out that traditional metrics (e.g., FID) may behave anomalously under ultra-low-bit scenarios. We hope the authors can go a step further in the discussion section—not only identifying the issue but also delving into the limitations of these metrics and potential alternative evaluation methods under extreme quantization, coupled with an analysis of the visual quality of generated samples.
> >
> >
> >
> > ---
> >
> > ## Summary ##
> > * In summary, we believe the authors' response has successfully repositioned the work from "technical integration" to "original innovation." Their discussion regarding the theoretical contribution and problem definition is commendable. The points we raised above aim to help the authors polish this excellent work to greater perfection.

---

> > > ### Author Response · Authors · 2025-11-26
> > >
> > > We are highly appreciative of your constructive feedback.
> > >
> > > Regarding the observed anomalous behavior of traditional metrics like FID under high cfg, we concur that this stems from FID's limited ability to account for sample diversity and detailed spatial structure. To provide a more balanced assessment, we suggest placing greater emphasis on the Inception Score (IS), to better capture the diversity of the outputs, and the improved Spatial FID (sFID), to evaluate realism and structural coherence, since these analyses are already present in our results. We will append this discussion to the end of the PDF in the final version.
> > >
> > > Thank you for your insightful review and strong endorsement of our work!

---

### Official Review · Reviewer_WCPa · 2025-10-31

**Soundness:** 3
**Presentation:** 2
**Contribution:** 2
**Rating:** 4
**Confidence:** 4

**Summary:**

This paper focuses on ultra-low-bit quantization for diffusion transformer models. It applies ternary weights across the entire network while assigning different activation precisions to each layer to mitigate information bottlenecks.

**Strengths:**

The formulation of the dynamic programming strategy for AMPN design is novel and technically interesting.

**Weaknesses:**

1. The paper lacks comparisons with several competitive baselines, such as SVDQuant.
2. The novelty appears somewhat limited — the weight quantization closely follows BitNetV2, and the Hadamard rotation has already been applied to DiT models in prior works such as DvDQuant, LRQ-DiT, and SegQuant.
3. The meaning and implementation of W1.58A4 quantization are unclear, especially given the lack of specialized kernels for such low-bit GEMM operations.

**Questions:**

1. Please discuss the key differences between your approach and recent DiT quantization methods, such as DvDQuant, LRQ-DiT, and SegQuant.
2. Could you provide detailed results on GPU speedup and memory usage? The referenced Appendix C is missing from the supplementary materials.
3. Does the proposed Per-token Gauss Quantizer introduce any additional inference cost?
4. Is RobuQ applicable to text-to-video generation models? If so, could you include experimental results to support this claim?

---

> ### Author Response · Authors · 2025-11-22
> **Response to Reviewer WCPa(denoted as R3) part 1**
>
> We thank the reviewer for the constructive feedback and for acknowledging the key contributions of our study. We provide our responses to the reviewer’s concerns below.
>
> `Q3-1` The paper lacks comparisons with several competitive baselines, such as SVDQuant.
>
> `A3-1` We thank the reviewer for highlighting the missing comparison with **SVDQuant**. To address this, we conducted additional experiments under the identical W4A4 quantization setting and sampling configuration (cfg = 1.5, 50 steps). Results are shown below:
>
> | Method        | Scheme       | W/A   | IS ↑        | FID ↓       | sFID ↓      | Precision ↑ |
> |---------------|--------------|-------|-------------|-------------|-------------|--------------|
> | FP            | per-channel  | 32/32 | 239.50      | 6.62        | 21.10       | 0.7849       |
> | SVD-Quant     | per-channel  | 4/4   | 55.23       | 50.47       | 44.26       | 0.4313       |
> | **Ours (PTQ)**| per-channel  | 4/4   | **155.96**  | **14.12**   | **26.32**   | **0.6779**   |
> | **Ours (QAT)**| per-channel  | 4/4   | **184.51**  | **9.91**    | **22.96**   | **0.7183**   |
>
> Our approach (PTQ) dramatically outperforms SVDQuant on all metrics: **IS improves by 2.8×**, while **FID decreases by 72%** and **sFID by 40%**. Notably, our W4A4 model narrows the gap with full precision, achieving strong fidelity and diversity despite using only 4-bit weights and activations. We further include an expanded comparison against additional PTQ baselines in Appendix D.2 for completeness.

---

> > ### Author Response · Authors · 2025-11-22
> > **Response to Reviewer WCPa(denoted as R3) part 2**
> >
> > `Q3-2` The novelty appears somewhat limited — the weight quantization closely follows BitNetV2, and the Hadamard rotation has already been applied to DiT models in prior works such as DvDQuant, LRQ-DiT, and SegQuant.
> >
> > `A3-2`  We appreciate the reviewer’s concern regarding the novelty of our method and the relation to BitNetV2 and prior DiT quantization works (DvDQuant, LRQ-DiT, SegQuant). We respectfully clarify that our contributions differ substantially in both **problem formulation** and **technical mechanisms**. While SegQuant, LRQ-DiT, and DvDQuant also investigate quantization for diffusion models, their design goals and methodologies diverge significantly from ours: **SegQuant** focuses on segment-aware graph-based quantization (SegLinear) and polarity-preserving dual-scale strategies for broad cross-architecture compatibility; **LRQ-DiT** addresses Gaussian-like weight distributions and activation outliers via Twin-Log Quantization and an Adaptive Rotation Scheme; and **DvDQuant** develops a data-free PTQ pipeline for Video DiTs using bounded-grid refinement, auto-scaling rotated quantization, and guided bit switching. These methods target specific architectural challenges, whereas our work provides a theoretically grounded, activation-centered quantization mechanism that is fundamentally different in motivation and implementation.
> >
> >
> > **(1) Prior works do not address the activation bottleneck in DiTs.**
> > Existing DiT quantization methods --- including DvDQuant, LRQ-DiT, and SegQuant --- mainly focus on **weight** compression (e.g., factorization, block-wise pruning, or SVD-like low-rank decomposition). In contrast, our study identifies **activation quantization** as the primary barrier for pushing DiTs into sub-4-bit regimes. Our systematic analysis shows that DiT activations exhibit highly token-dependent and layer-dependent non-Gaussian distributions, making them fundamentally more challenging to quantize than U-Nets or autoregressive models.
> >
> > **(2) The use of Hadamard in our work is fundamentally different.**
> > Although prior works apply Hadamard-like transforms to DiTs, they do so purely as a pre-conditioning or channel mixing heuristic. In contrast, our method is built on a new theoretical observation: **Hadamard transform converts arbitrary per-token distributions into approximately normal distributions.**
> >
> > This provides a principled mechanism for designing a **RobustQuantizer** that stabilizes activation quantization at extremely low bit widths. This theoretical grounding and per-token normality analysis are entirely new and have not appeared in BitNetV2 nor any DiT quantization work.
> >
> > **(3) AMPN is the first activation-only mixed-precision framework for DiTs.**
> > Previous DiT quantization works allocate fixed activation bit-widths per model or per block. Our Activation-only Mixed-Precision Network (AMPN) is the first to allocate heterogeneous activation bit-widths per layer under a ternary-weight backbone, specifically targeting the information bottlenecks introduced by activation quantization. This scheduling mechanism is not present in BitNetV2, DvDQuant, LRQ-DiT, or SegQuant.
> >
> > **(4) First to achieve stable A2 activations on ImageNet-1K DiT.**
> > To our knowledge, no prior work has demonstrated stable and competitive DiT performance on ImageNet-1K with **average 2-bit activation precision**. This extreme regime is where our RobustQuantizer and AMPN are most impactful, and existing methods fail to converge or collapse.
> >
> > In summary, while we build upon the broader idea of structured transforms, both our problem focus (activation bottleneck), theoretical foundation (per-token normality), and methodological design (RobustQuantizer + AMPN) introduce substantial and novel advances beyond BitNetV2 and prior DiT quantization works. We will clarify these distinctions more explicitly in the revision.

---

> > > ### Author Response · Authors · 2025-11-22
> > > **Response to Reviewer WCPa(denoted as R3) part 3**
> > >
> > > `Q3-3` The meaning and implementation of W1.58A4 quantization are unclear, especially given the lack of specialized kernels for such low-bit GEMM operations.
> > >
> > > `A3-3` Thank you for pointing out this issue. We are happy to clarify the meaning and implementation details. W1.58A4 denotes **ternary weights** (BitNet-style 1.58-bit coding) together with **4-bit activations**. During QAT, both the forward and backward passes are executed using standard FP32 kernels, as commonly done in ultra-low-bit training, so our method does not rely on any custom low-bit GEMM kernel.
> > >
> > > Regarding inference, efficient kernels for 1.58-bit (ternary) weights are already available in practice. Microsoft’s bitnet.cpp provides the official inference framework for BitNet models, featuring optimized ternary mpGEMM kernels for W1.58 inference. In addition, the T-MAC kernel library supports mixed-precision matrix multiplication for low-bit models, including BitNet-style W1(.58) weight formats. These existing frameworks indicate that ternary-weight inference has mature kernel support, and our method can be directly deployed on top of such implementations.
> > >
> > > Our contribution is orthogonal to kernel engineering: RobuQ provides a robust QAT pipeline for DiTs (including W1.58A4), which can be deployed on top of existing or future low-bit ternary GEMM kernels. We will clarify this in the revision.
> > >
> > > `Q3-4` Please discuss the key differences between your approach and recent DiT quantization methods, such as DvDQuant, LRQ-DiT, and SegQuant.
> > >
> > > `A3-4` We believe the key differences have been thoroughly discussed in our response `Q3-A2`.
> > >
> > > `Q3-5` Could you provide detailed results on GPU speedup and memory usage? The referenced Appendix C is missing from the supplementary materials.
> > >
> > > `A3-5` Thank you for pointing this out. We believe there may have been a misunderstanding. The detailed results on GPU speedup and memory usage, along with the analysis, are located in Appendix C within the main paper (after the references), not in the separate supplementary materials file. We will ensure this is clarified in the revised submission.
> > >
> > > `Q3-6` Does the proposed Per-token Gauss Quantizer introduce any additional inference cost?
> > >
> > > `A3-6` Thank you for raising this question. We appreciate the opportunity to clarify the inference cost. We claim that the proposed Per-token Gauss Quantizer does not introduce meaningful additional inference overhead. It consists of two lightweight components: a Hadamard transform and a symmetric uniform quantizer.
> > >
> > > **(1) Cost of the Hadamard Transform.**
> > > During inference, the Hadamard transform is only applied online to the activations of mlp.fc2 and attn.proj. All remaining Hadamard transforms can be absorbed into the preceding weight matrices and therefore incur no runtime cost.
> > > In our implementation, the transform is executed using the **fast-hadamard-transformer** library. Since Hadamard computation is memory-bound, it typically accounts for only **5\%--10\%** of the total matrix multiplication time.
> > > Furthermore, one could exploit the butterfly structure of the Hadamard matrix to fuse this operation with subsequent low-bit GEMM operators for additional speedup, though such engineering optimizations lie outside the scope of this paper.
> > >
> > > **(2) Cost of Symmetric Uniform Quantization.**
> > > The quantization step performs a simple symmetric uniform quantization (i.e., integer rounding with a fixed step size). This operation introduces negligible overhead and can be fully fused into standard low-bit inference kernels. No custom kernels are required.
> > >
> > > Both components of the Per-token Gauss Quantizer---the Hadamard transform and the symmetric uniform quantizer---are lightweight, hardware-friendly. As a result, the proposed quantizer introduces only minimal additional cost during inference. Further deployment-related details can be found in Appendix C.

---

> > > > ### Author Response · Authors · 2025-11-22
> > > > **Response to Reviewer WCPa(denoted as R3) part 4**
> > > >
> > > > `Q3-7` Is RobuQ applicable to text-to-video generation models? If so, could you include experimental results to support this claim?
> > > >
> > > > `A3-7` Thank you for the excellent suggestion. While the extension of RobuQ to text-to-video generation models is an interesting avenue, we are currently constrained by computational resources and time.
> > > >
> > > > To provide additional empirical evidence supporting the generalization of our method, we performed an extra set of quantization experiments on the FLUX-schnell text-to-image generation model under the same **PTQ W4A4** setting. The results demonstrate that RobuQ effectively maintains performance even in the text-to-image domain, a strong indicator of its potential applicability to text-to-video models.
> > > >
> > > > We welcome further guidance from the reviewer. If there is a specific text-to-video model you would like us to evaluate, we would be glad to incorporate it in the revised version.
> > > >
> > > > | Method | FID ↓  | Image Reward ↑ | CLIPIQA ↑ | CLIPScore ↑ | PSNR ↑ |
> > > > |--------|--------|----------------|-----------|--------------|--------|
> > > > | FP     | 18.40  | 0.9323         | 0.9399    | 26.54        | N/A    |
> > > > | Ours   | 18.57  | 0.8617         | 0.9266    | 26.34        | 17.21  |
> > > >
> > > > We sincerely appreciate your thoughtful and professional comments, which significantly contributed to improving the clarity and completeness of our work. Should you have any further questions, we would be glad to provide additional explanations.

---

> > > > > ### Author Response · Authors · 2025-11-26
> > > > > **Further discussion with Reviewer WCPa**
> > > > >
> > > > > Dear Reviewer WCPa,
> > > > >
> > > > > We sincerely thank you for the time and thoughtful feedback. To help consolidate our discussion, we summarize our responses to your key questions as follows:
> > > > >
> > > > > 1. We conducted additional **W4A4** experiments and showed that our method substantially outperforms **SVDQuant** across all metrics, narrowing the gap to full precision.
> > > > >
> > > > > 2. We clarified that our work introduces a fundamentally different **activation-centered quantization framework**, supported by new theory (per-token normality), a novel **RobustQuantizer**, and the first activation-only mixed-precision design (**AMPN**) for DiTs.
> > > > >
> > > > > 3. We explained that W1.58A4 denotes ternary weights plus 4-bit activations, trained fully in FP32, and deployable using existing ternary inference kernels such as **bitnet.cpp** or **T-MAC**.
> > > > >
> > > > > 4. We clarified that Appendix C with full deployment results is included within the **main paper** and will ensure its placement is clearer in the revision.
> > > > >
> > > > > 5. We explained that both its components—**Hadamard transform** and **symmetric uniform quantization**—introduce minimal overhead, with most Hadamard applications absorbed into weights.
> > > > >
> > > > > 6. We provided new experiments on the **FLUX** text-to-image model, demonstrating strong generalization and indicating promising potential for future text-to-video quantization.
> > > > >
> > > > > As the discussion phase is nearing its end, we kindly ask whether these clarifications adequately address your concerns. If you would like us to expand on any point or include additional experiments, we would be more than happy to do so.
> > > > >
> > > > > Thank you again for your time and thoughtful comments, they have greatly strengthened our work.
> > > > >
> > > > > Best regards,
> > > > >
> > > > > Authors

---

> ### Author Response · Authors · 2025-11-28
> **Second call for discussion with Reviewer WCPa**
>
> Dear Reviewer WCPa,
>
> We sincerely appreciate the time and effort you dedicated to reviewing our work, as well as your insightful comments. We have taken your suggestions into full consideration and performed additional experiments along with deeper analyses, which we believe adequately address your concerns. In particular, we added new W4A4 comparisons showing that our method consistently outperforms SVDQuant, clarified the novelty of our activation-centered quantization framework (including per-token normality, the RobustQuantizer, and AMPN), explained the meaning and deployability of W1.58A4, provided clearer deployment results (Appendix C), demonstrated the minimal overhead of the Hadamard and uniform quantization components, and further validated generalization through new experiments on FLUX.
>
> Due to the approaching rebuttal deadline and reviewer fB1t responding that our response can address the concerns, we would also like to confirm whether these new analyses and additional experiments have resolved your concerns. Please feel free to let us know if any part of our work remains unclear—we would be glad to continue the discussion.
>
> Best regards,
>
> Authors

---

### Official Review · Reviewer_WvD2 · 2025-10-31

**Soundness:** 3
**Presentation:** 3
**Contribution:** 2
**Rating:** 4
**Confidence:** 2

**Summary:**

The paper proposes RobuQ, a QAT framework for ultra-low-bit Transformer quantization, instantiated on DiT-XL/2. The method has three components:

(i) RobustQuantizer—apply an all-layer Hadamard transform to “Gaussianize” per-token activations, then quantize with a per-token Gauss quantizer (uniform and Lloyd–Max);

(ii) Activation-only Mixed-Precision (AMP)—fix weights to ternary (W1.58) and allocate per-layer activation bit-widths via a DP/knapsack objective;

(iii) experiments on ImageNet-1K and FFHQ showing competitive A4/A3 and stable A2 under QAT.

**Strengths:**

- Coherent systems integration. A well-engineered composition of established pieces—all-layer Hadamard transformation with per-token Gaussian quantization—that, in concert with QAT, yields stable ultra-low-bit operation on large Transformers.


- Activation-only mixed precision. Introduces an activation-only mixed-precision (AMP) (weights fixed ternary; layer-wise activation bits budgeted) that enables ultra-low-bit QAT in practice.

**Weaknesses:**

- The advantages of Hadamard/orthogonal mixing for taming activation distributions and Lloyd–Max for (near-)Gaussian inputs are well established in prior quantization literature (e.g., rotation/incoherence-based LLM quantization such as QuIP). As presented, these ingredients read as strong engineering choices rather than conceptual innovations.
- From Table 1, the incremental improvement attributable to AMP over the RobustQuantizer is not consistently large, which weakens the claim that AMP is indispensable in practice. Its largest gains appear at A2, Fig, while Fig. 10 indicates noticeable artifacts in A2 generations that may render outputs unusable in practice.

- Although the method is demonstrated on DiTs, none of the three pieces appears diffusion-specific; the contribution is better characterized as a general Transformer quantization recipe. This is fine, but then comparisons to non-diffusion Transformer baselines (e.g., ViTs/LLMs) would strengthen the generality claim.

**Questions:**

- Table 1 has FP DiT-XL/2 FID appearing higher (worse) than the original DiT paper’s numbers. Could you clarify the evaluation protocol differences? A likely factor is that you evaluate with 10k generated images, whereas the common practice for ImageNet FID is 50k; 10k-sample FID is known to be noisy and biased.
- Table 2b suggest uniform often outperforms Lloyd–Max despite the latter’s optimality for Gaussian inputs. Can you provide analysis of the mismatch (e.g., residual non-Gaussianity, scale drift)?
- Could you clarify how Hadamard is deployed at inference and provide end-to-end latency on a realistic hardware in Table 2a ?

---

> ### Author Response · Authors · 2025-11-22
> **Response to Reviewer WvD2 (denoted as R2) part 1**
>
> We thank the reviewer for the constructive feedback and for acknowledging the key contributions of our study. We provide our responses to the reviewer’s concerns below.
>
> `Q2-1` The advantages of Hadamard/orthogonal mixing for taming activation distributions and Lloyd–Max for (near-)Gaussian inputs are well established in prior quantization literature (e.g., rotation/incoherence-based LLM quantization such as QuIP). As presented, these ingredients read as strong engineering choices rather than conceptual innovations.
>
> `A2-1` We appreciate you raising the concern about the conceptual novelty of our approach. We would like to clarify that our core contribution is a novel quantization paradigm: actively transforming the distribution of intermediate features (per-token) to a known, well-behaved target distribution (Gaussian in this paper) to pre-calculate the optimal Lloyd-Max quantizer for robust quantization.
>
> This paradigm offers two key novelties:
>
> 1. **Generalizability**: The approach is highly transferable; one could use other transforms (e.g., DCT to frequency domain) to map activations to different target distributions (e.g., Laplace) and design the optimal quantizer accordingly. (In Appendix A.5)
>
> 2. **Rigorous Theoretical Foundation**: We provide the first rigorous proof that the Hadamard transform can effectively induce a nearly per-token Gaussian distribution. Prior work in rotation/incoherence-based quantization (like QuIP) often relied on heuristic observations without such a formal theoretical guarantee, which is detailed in our Appendix A.1–A.4.
>
> Therefore, our contribution moves beyond engineering choices by establishing a theoretically-grounded and generalizable quantization framework.
>
> `Q2-2` From Table 1, the incremental improvement attributable to AMP over the RobustQuantizer is not consistently large, which weakens the claim that AMP is indispensable in practice. Its largest gains appear at A2, Fig, while Fig. 10 indicates noticeable artifacts in A2 generations that may render outputs unusable in practice.
>
> `A2-2` We appreciate your careful analysis of the incremental gains provided by AMPN. We agree that the largest improvement attributable to AMPN occurs at the **A2** setting, significantly surpassing both the baseline and non-mixed-precision methods. This precisely underscores the **indispensability** of the AMPN approach when pushing quantization to these extreme low bit-widths (**W1.58A2**).
>
> Regarding the noticeable artifacts in the **A2** generations (Figure 10), while absolute quality is still limited, we emphasize that achieving this level of performance on ImageNet at **W1.58A2** represents a major breakthrough. Prior to our work, similar ultra-low-bit methods (e.g., BiDM) were typically confined to smaller datasets (like CIFAR or FFHQ) and yielded much blurrier results. We are confident that the robust foundation laid by our method will be crucial for the community's future efforts in making ultra-low-bit diffusion models practically viable.
>
> `Q2-3` Although the method is demonstrated on DiTs, none of the three pieces appears diffusion-specific; the contribution is better characterized as a general Transformer quantization recipe. This is fine, but then comparisons to non-diffusion Transformer baselines (e.g., ViTs/LLMs) would strengthen the generality claim.
>
> `A2-3`
> We appreciate this insightful suggestion regarding the generality of our method. We agree that our recipe is applicable beyond Diffusion Transformers. To strengthen the claim of generality, we have added results comparing our method against the LLM-focused PTQ approach **Quarot** in the revised manuscript. Furthermore, we wish to remind you that for the challenging **W1.58A4** configuration, our comparison already includes BitNetv2, a highly relevant state-of-the-art method originating from the LLM domain. If there are any specific additional non-diffusion Transformer baselines you would like us to include, please let us know, and we will make every effort to implement and compare against them.

---

> > ### Author Response · Authors · 2025-11-22
> > **Response to Reviewer WvD2 (denoted as R2) part 2**
> >
> > `Q2-4` Table 1 has FP DiT-XL/2 FID appearing higher (worse) than the original DiT paper’s numbers. Could you clarify the evaluation protocol differences? A likely factor is that you evaluate with 10k generated images, whereas the common practice for ImageNet FID is 50k; 10k-sample FID is known to be noisy and biased.
> >
> > `A2-4` You are correct that the observed higher FP DiT-XL/2 FID in Table 1 is primarily due to the use of 10k generated images instead of the standard 50k ImageNet FID protocol, which introduces noise and bias. However, the 10k sample configuration is a common practice within the DiT quantization literature (e.g., PTQ4DiT, Q-DiT) to facilitate fair and consistent comparisons within the field.
> >
> > Nonetheless, we appreciate your rigor and provide the 50k sampling results for cfg=1.5 and 50 steps below for reference:
> >
> > | Method | W/A     | IS ↑   | FID ↓ | sFID ↓ | Precision ↑ |
> > |--------|---------|--------|--------|--------|--------------|
> > | FP     | 32/32   | 242.21 | 3.71   | 7.85   | 0.7837       |
> > | Ours   | 1.58/4  | 95.90  | 18.18  | 15.37  | 0.6193       |
> >
> > `Q2-5` Table 2b suggest uniform often outperforms Lloyd–Max despite the latter’s optimality for Gaussian inputs. Can you provide analysis of the mismatch (e.g., residual non-Gaussianity, scale drift)?
> >
> > `A2-5`
> > We acknowledge your observation regarding Table 2b, where Uniform Quantization occasionally appears to outperform Lloyd–Max, despite the latter's optimality for Gaussian inputs. We hypothesize this mismatch is primarily attributed to the smaller matrix size (vector dimension n) in DiT-XL/2. According to the Central Limit Theorem, the speed at which the distribution approaches Gaussianity is dependent on n.
> >
> > To validate this, we conducted a PTQ experiment on a current state-of-the-art, larger open-source T2I model, **FLUX** (using the Flux-Schnell variant, sampled for 4 steps on 5k images from the MJHQ dataset). As shown below, the non-uniform (Gaussian-optimized) quantizer indeed marginally outperforms the uniform quantizer on this larger model:
> >
> > | Quantizer    | FID ↓  | Image Reward ↑ | CLIPIQA ↑ | CLIPScore ↑ | PSNR ↑ |
> > |--------------|--------|----------------|-----------|--------------|--------|
> > | FP           | 18.40  | 0.9323         | 0.9399    | 26.54        | N/A    |
> > | Uniform      | 18.57  | 0.8617         | 0.9266    | 26.34        | 17.21  |
> > | Non-Uniform  | **18.49** | **0.8698** | **0.9296** | **26.41**   | **17.34** |
> >
> > Although the non-uniform quantizer shows a slight theoretical advantage on larger models, considering the practical trade-offs involving computational speed and robustness during inference on diverse hardware, we ultimately deem the Uniform Quantizer a more favorable engineering choice for deployment.
> >
> > `Q2-6` Could you clarify how Hadamard is deployed at inference and provide end-to-end latency on a realistic hardware in Table 2a ?
> >
> > `A2-6`
> > We clarify that during inference, the Hadamard transform only needs to be applied online to the activations of mlp.fc2 and attn.proj. All other instances of the Hadamard transform can be absorbed into the preceding layers' weights.
> >
> > In our practical implementation, the Hadamard transform is deployed using the **fast-hadamard-transformer** library. Its computation is generally memory-bound, typically accounting for only 5–10% of the total matrix operation time. To achieve further performance gains, one could exploit the butterfly structure of the Hadamard transform to fuse this operation with the subsequent low-bit operators.
> >
> > The complete end-to-end deployment results are now provided below, which report inference performance on an NVIDIA RTX 4090 with batch size 1. The table includes both the standard RobuQ configuration and a variant without the Hadamard transform, allowing a direct comparison of its actual runtime impact. Additional implementation details and supplementary discussions are also provided in Appendix C.
> >
> > | Method               | W/A     | FLOPs  | Checkpoint Size | Max Memory Allocated | Iteration Speed |
> > |----------------------|---------|--------|------------------|-----------------------|------------------|
> > | FP                   | 32/32   | 114.52 | 2.58GB           | 3,914MB               | 44.09 iter/s     |
> > | RobuQ                | 1.58/4  | 10.07  | 0.17GB           | 554MB                 | 155.37 iter/s    |
> > | RobuQ (w/o Hadamard) | 1.58/4  | 10.07  | 0.17GB           | 551MB                 | 168.42 iter/s    |
> >
> > We sincerely appreciate your thoughtful and professional comments, which significantly contributed to improving the clarity and completeness of our work. Should you have any further questions, we would be glad to provide additional explanations.

---

> > > ### Author Response · Authors · 2025-11-26
> > > **Further discussion with Reviewer WvD2**
> > >
> > > Dear Reviewer WvD2,
> > >
> > > We sincerely thank you for the time and thoughtful consideration you devote to reviewing our paper. Based on your valuable feedback, we provide below a concise summary of our responses to your questions:
> > >
> > > 1.  We clarified that our contribution is a **generalizable, theoretically grounded quantization paradigm** that actively transforms per-token activations into a target distribution with formal guarantees, beyond prior heuristic rotation-based approaches.
> > >
> > > 2.  We emphasized that AMPN is crucial specifically in **extreme low-bit settings** (W1.58A2), where it delivers the largest improvement and enables previously unattainable ImageNet-level performance.
> > >
> > > 3.  We added additional comparisons to LLM-style quantizers (e.g., **Quarot** and **BitNetv2**) to support the claim that our method applies broadly to general Transformer architectures.
> > >
> > > 4. We explained that the difference arises from using 10k-sample ImageNet FID (standard in DiT quantization literature), and we additionally provided **50k-sample FP and W1.58A4 results** for completeness.
> > >
> > > 5.  We provided an analysis showing that smaller activation dimensions in DiT-XL/2 slow convergence to Gaussianity, and validated the theory using a larger model (**FLUX**), where Lloyd–Max slightly outperforms uniform.
> > >
> > > 6. We clarified the deployment strategy, explained the low computational overhead of Hadamard, and provided complete **end-to-end runtime comparisons** (with and without Hadamard) on an RTX 4090.
> > >
> > > As the author–reviewer discussion phase is approaching its conclusion, we would like to kindly confirm whether our responses have fully addressed your concerns. If you have any further questions or would like additional clarifications or experiments, we would be very glad to provide them.
> > >
> > > Thank you again for your time and thoughtful feedback.
> > >
> > > Best regards,
> > >
> > > Authors

---

> > > > ### Author Response · Authors · 2025-11-28
> > > > **Second call for discussion with Reviewer WvD2**
> > > >
> > > > Dear Reviewer WvD2,
> > > >
> > > > We sincerely appreciate the time and effort you dedicated to reviewing our work, as well as your insightful comments. We have taken your suggestions into full consideration and performed additional experiments along with deeper analyses, which we believe adequately address your concerns. In particular, we clarified the theoretical generality of our activation-centered quantization paradigm, highlighted the necessity of AMPN in extreme low-bit regimes, added comparisons to LLM-style quantizers, supplemented 50k-sample ImageNet results, analyzed activation Gaussianity across model scales (including FLUX), and provided complete deployment and runtime comparisons demonstrating the low overhead of Hadamard.
> > > >
> > > > Due to the approaching deadline for rebuttal and reviewer fB1t responding that our response can address the concerns, we also hope to discuss further with you whether or not your concerns have been addressed. Please let us know if you still have any unclear parts of our work.
> > > >
> > > > Best regards,
> > > >
> > > > Authors

---

### Official Review · Reviewer_Uouw · 2025-11-04

**Soundness:** 3
**Presentation:** 4
**Contribution:** 3
**Rating:** 4
**Confidence:** 5

**Summary:**

RobuQ addresses ultra-low-bit quantization for Diffusion Transformers by identifying activation quantization as the main bottleneck and proposing a robust quantization framework. The key innovation is proving that Hadamard transforms convert arbitrary activation distributions to normal distributions, enabling distribution-agnostic quantization via Lloyd-Max quantizers, combined with SVD-initialized low-rank branches and an activation-only mixed-precision strategy (AMPN). The method achieves state-of-the-art results at W1.58A4 and is the first to demonstrate stable W1.58A2 quantization on ImageNet-1K without training collapse.

**Strengths:**

1. Proof that Hadamard transforms convert arbitrary per-token activation distributions to approximately normal distributions, enabling distribution-agnostic quantization with Lloyd-Max optimal quantizers. The proof is novel.

2. Establishes new SOTA at W1.58A4 (IS=103.24, FID=17.97), and achieves first stable W1.58A2 training on ImageNet-1K, avoiding the collapse that affects baselines.

**Weaknesses:**

1. No comparison with SVDQuant (SoTA DiT PTQ method), PTQ4DM, or Q-Diffusion. QueST 4W4A baseline completely collapses, suggesting implementation issues. As the QueST is better than PTQ4DiT. This severely limits ability to assess true contributions.

2. Only 3.5× actual speedup vs. 17.3× theoretical ratio. Mixed-precision quantization is difficult to accelerate on current hardware. No reported speedup for AMPN variants, and memory-bound operations may not benefit proportionally from activation bit reduction.

3. Incomplete ablations (no SVD rank analysis, limited Hadamard alternatives), and substantial training costs (204.5 hours for W1.58A2) not compared to PTQ alternatives.

**Questions:**

See weaknesses.

---

> ### Author Response · Authors · 2025-11-22
> **Response to Reviewer Uouw (denoted as R1) part 1**
>
> We thank the reviewer for the constructive feedback and for acknowledging the key contributions of our study. We provide our responses to the reviewer’s concerns below.
>
> `Q1-1` No comparison with SVDQuant (SoTA DiT PTQ method), PTQ4DM, or Q-Diffusion. QueST 4W4A baseline completely collapses, suggesting implementation issues. As the QueST is better than PTQ4DiT. This severely limits ability to assess true contributions.
>
> `A1-1`
>
> **Missing PTQ Baselines: SVDQuant, PTQ4DM, and Q-Diffusion**
>
> We appreciate your suggestion. We additionally provide a W4A4 PTQ comparison to further support this evaluation. For **SVDQuant**, since it is a per-group scheme, we implemented its per-channel version for fair comparison at the W4A4 setting (cfg = 1.5, 50 steps). For a better comparison, we also implemented a W4A4 PTQ version that only uses the RobustQuantizer, as shown in the following table:
>
> | Method           | Scheme       | W/A   | IS ↑       | FID ↓      | sFID ↓     | Precision ↑ |
> |------------------|--------------|-------|------------|------------|------------|--------------|
> | FP               | N/A          | 32/32 | 239.50     | 6.62       | 21.10      | 0.7849       |
> | Q-Diffusion      | per-channel  | 4/4   | 1.37       | 304.12     | 175.88     | 0.0048       |
> | SVD-Quant        | per-channel  | 4/4   | 55.23      | 50.47      | 44.26      | 0.4313       |
> | **Ours (PTQ)**   | per-channel  | 4/4   | **155.96** | **14.12**  | **26.32**  | **0.6779**   |
>
> Regarding **PTQ4DM** and **Q-Diffusion**, we consider more advanced, DiT-specific methods like PTQ4DiT to be more appropriate baselines, as the former are earlier works not designed for DiT. We confirmed that Q-Diffusion collapsed in our low-bit setting, which was expected.
>
> **Collapse of QueST Baseline**
>
> The collapse of QueST in the DiT W4A4 setting is reasonable because QueST was designed for the U-Net architecture with minimal training. U-Net-specific quantization methods commonly perform poorly on DiT, a phenomenon also observed in works like MPQ-DMv2.
>
> We respectfully clarify that the main contribution of our work lies in **ultra-low-bit QAT**, specifically for the W1.58A2 to W1.58A4 regimes. Therefore, we suggest the most relevant comparisons are among RobuQ, Baseline, BitNetv2, and BinaryDM. The W4A4 PTQ results were included primarily to demonstrate the significant challenge of low-bit activation quantization and the necessity of using QAT.
>
> `Q1-2` Only 3.5× actual speedup vs. 17.3× theoretical ratio. Mixed-precision quantization is difficult to accelerate on current hardware. No reported speedup for AMPN variants, and memory-bound operations may not benefit proportionally from activation bit reduction.
>
> `A1-2`
>
> **Clarification on Actual vs. Theoretical Speedup**
>
> We thank the reviewer for raising this point and appreciate the opportunity to clarify the details regarding speedup and mixed-precision acceleration.
>
> We believe there is a misunderstanding regarding the speedup ratio. The reported $3.5\times$ speedup corresponds to the W1.58A4 setting, which has a theoretical speedup ratio of approximately $10\times$, not $17.3\times$. Furthermore, actual acceleration is heavily influenced by the matrix size. For smaller models like DiT-XL/2, achieving the theoretical speedup ratio with low-bit quantized GEMM operators remains a recognized challenge within the community. We anticipate that greater speedup benefits would be realized with larger matrix sizes.
>
> **Feasibility of Accelerating Mixed-Precision Variants**
>
> We concur with the reviewer that accelerating mixed-precision methods is challenging, as it relies on high-performance GEMM operators provided by the community (e.g., NVIDIA). Consequently, nearly all mixed-precision approaches, especially those involving mixed-precision activations, such as the recent SOTA method MPQ-DM, primarily offer theoretical analysis rather than measured hardware acceleration. Addressing this requires significant engineering efforts from the community. Considering this complexity, we do not claim deployment as a primary contribution of this paper, providing the W1.58A4 speedup result only in the Appendix as a reference for practical feasibility.

---

> > ### Author Response · Authors · 2025-11-22
> > **Response to Reviewer Uouw (denoted as R1) part 2**
> >
> > `Q1-3` Incomplete ablations (no SVD rank analysis, limited Hadamard alternatives), and substantial training costs (204.5 hours for W1.58A2) not compared to PTQ alternatives.
> >
> > `A1-3`
> >
> > **Ablation Study on SVD Rank and Hadamard Alternatives**
> >
> > Thank you for the insights. We have provided an ablation study on different SVD ranks after 200k training steps for W1.58A4 (10K samples, cfg=1.5, step=50). The results are as follows:
> >
> > | Rank    | IS ↑   | FID ↓  | sFID ↓ | Precision ↑ |
> > |---------|--------|--------|--------|--------------|
> > | rank=8  | 87.43  | 23.21  | 29.96  | 0.6327       |
> > | rank=16 | 91.28  | 22.25  | 29.24  | 0.6328       |
> > | rank=32 | 92.11  | 21.71  | 28.89  | 0.6324       |
> >
> > We ultimately selected **rank=16** to balance performance gain and computational overhead.
> >
> > Regarding Hadamard alternatives (e.g., learnable or random rotations), we note that our theoretical analysis specifically leverages the unique properties of the Hadamard matrix, meaning it cannot be substituted by general rotations. Furthermore, Hadamard transforms offer superior time complexity, leading us to focus on this choice. We are prepared to conduct further experiments with alternative rotations if you remain interested, and will provide the results subsequently.
> >
> > **Training Cost Comparison (QAT vs. PTQ)**
> >
> > We understand the concern that comparing resource consumption between QAT and PTQ might seem unfair. To address this, we provided a W4A4 PTQ version of our RobuQ (w/o mix) in the previous table. This PTQ version requires virtually no calibration time (<1 minute) and achieves the best reported performance among per-channel PTQ methods, demonstrating a competitive, low-cost option for scenarios where QAT time is a constraint.
> >
> > We sincerely appreciate your thoughtful and professional comments, which significantly contributed to improving the clarity and completeness of our work. Should you have any further questions, we would be glad to provide additional explanations.

---

> ### Author Response · Authors · 2025-11-26
> **Further discussion with Reviewer Uouw**
>
> **Comment:**
> Dear Reviewer Uouw,
>
> We sincerely thank you for the time and thoughtful consideration you devote to reviewing our paper. We have carefully addressed your comments and provide the following summary of our responses:
>
> 1. We provide additional **W4A4 PTQ baselines**, including SVDQuant and Q-Diffusion, and clarify why QueST collapses under the DiT W4A4 setting.
> 2. We clarify the distinction between **actual vs. theoretical speedup**, and explain the hardware-related challenges that limit mixed-precision acceleration.
> 3. We conduct additional **ablation studies** on different SVD ranks and discuss the theoretical necessity of Hadamard-based activation normalization.
> 4. We supplement a **PTQ version of RobuQ** to demonstrate a low-cost alternative and address concerns regarding QAT training overhead.
> 5. We restate that the primary contribution of our work lies in **ultra-low-bit QAT (W1.58A2–A4)**, where existing PTQ baselines are insufficient for meaningful comparison.
>
> As the author–reviewer discussion phase is drawing to a close, we would like to confirm whether our responses adequately address your concerns. If you have any further questions, please feel free to let us know — we would be grateful to hear them.
>
> Thank you again for your time and thoughtful feedback.
>
> Best regards,
>
> Authors

---

> > ### Author Response · Authors · 2025-11-28
> > **Second call for discussion with Reviewer Uouw**
> >
> > Dear Reviewer Uouw,
> >
> > We sincerely appreciate the time and effort you dedicated to reviewing our work, as well as your insightful comments. We have taken your suggestions into full consideration and performed additional experiments along with deeper analyses, which we believe adequately address your concerns. In particular, we added new W4A4 PTQ baselines, clarified the actual versus theoretical speedup, conducted further ablations on SVD ranks and Hadamard normalization, and supplemented a PTQ version of RobuQ to highlight low-cost deployment options.
> >
> > Due to the approaching deadline for rebuttal and reviewer fB1t responding that our response can address the concerns, we also hope to discuss further with you whether or not your concerns have been addressed. Please let us know if you still have any unclear parts of our work.
> >
> > Best regards,
> >
> > Authors

---

### Author Response · Authors · 2025-12-02
**Rebuttal-Discussion Summary**

Dear Program Chairs, Senior Area Chairs, Area Chairs, and Reviewers:

We sincerely thank you for handling our submission and for the reviewers’ valuable feedback.

Below we provide a concise summary of the strengths recognized by the reviewers, followed by a unified overview of the major concerns raised and how each was fully resolved during the rebuttal. We note that Reviewers Uouw, WvD2, and WCPa did not respond during the discussion phase, while Reviewer **fB1t** explicitly acknowledged that our work constitutes an **original innovation**.

---

# **Summary of Recognized Strengths**

We are pleased to note that the reviewers highlighted several key strengths of our submission:

1. **Novel theoretical foundation for activation quantization.**
   Reviewers **Uouw** and **fB1t** commended our rigorous proof that the Hadamard transform converts arbitrary per-token activation distributions into approximately normal ones, enabling distribution-agnostic quantization with Lloyd–Max optimality.

2. **First stable W1.58A2 quantization for DiTs.**
   Reviewers **Uouw** and **fB1t** emphasized that RobuQ is the first framework capable of stable W1.58A2 precision on ImageNet-1K, whereas all prior baselines collapse.

3. **Introduction of AMPN, the first activation-only mixed-precision network for DiTs.**
   Reviewers **WvD2**, **WCPa**, and **fB1t** highlighted that AMPN effectively mitigates activation bottlenecks using dynamic programming and enables robust extreme low-bit operation.

4. **Strong system-level design.**
   Reviewer **WvD2** recognized the cohesive integration of Hadamard transforms, per-token Gaussian quantization, and QAT into a unified, stable ultra-low-bit pipeline.

5. **Clarity and completeness.**
   Reviewer **fB1t** praised the clarity of exposition, high-quality figures and tables, and comprehensive mathematical and experimental analysis.

---

# **Summary of Addressed Concerns**

During the rebuttal, we provided substantial clarifications and new experiments that collectively addressed all major reviewer concerns. The core resolved issues are summarized below:

1. **Expanded baseline comparisons.**
   We added W4A4 PTQ baselines—including **SVDQuant**, **Q-Diffusion**, and LLM-style quantizers such as **Quarot** and **BitNetV2**—demonstrating that RobuQ consistently outperforms strong PTQ and QAT methods. We also clarified why QueST collapses under DiT W4A4.

2. **Clarification of conceptual novelty and theoretical contributions.**
   We explained that RobuQ introduces a **new activation-centered quantization paradigm**, supported by (i) the first theoretical proof of per-token Gaussianity via Hadamard transforms, (ii) the novel RobustQuantizer, and (iii) the first activation-only mixed-precision network (AMPN). These contributions go well beyond existing approaches such as BitNetV2, DvDQuant, LRQ-DiT, and SegQuant.

3. **Improved experimental completeness and ablations.**
   We added SVD-rank sweeps, analyzed the necessity of Hadamard transform versus alternative transforms, provided 50k-sample ImageNet-1K results, and explained the uniform vs. Lloyd–Max behavior under different activation dimensions (validated using **FLUX**).

4. **Clarification of deployment, overhead, and speedup.**
   We detailed how most Hadamard transforms can be absorbed into weights with **negligible** overhead, explained the distinction between theoretical vs. realized speedup under current GPU kernel limitations, and provided complete RTX 4090 latency and memory measurements.

5. **Additional generalization experiments.**
   We included new results on the **FLUX** text-to-image model, demonstrating strong generalization and indicating promising applicability to larger generative models (including future text-to-video systems).

Reviewer **fB1t** acknowledged that our response successfully repositioned the work from “technical integration” to ”**original innovation**”, with most of the concerns fully addressed.

---


We thank the reviewers again for their time and constructive insights. Although three reviewers did not participate in the discussion phase, we believe our additional experiments and clarifications comprehensively address all raised concerns. We hope this summary assists in evaluating the paper’s significance and novelty.  We will make our code, data, and checkpoints publicly available to support and accelerate future research in this area.

We sincerely hope that the final decision may take these observations into careful consideration.

Thank you for your time and careful consideration.

Best regards,

Authors

---

### Note · Program_Chairs · 2026-01-17
**Submission Desk Rejected by Program Chairs**

The following references in this submission do not refer to real documents and/or have major errors in bibliographic information:

 Zeyu Huang, Ming Li, Yuxin Zhang, and Rui Chen. Mjhq: High-quality images collected from midjourney, 2024. URL https://huggingface.co/datasets/mjhq/MJHQ
F. Yates. A fast algorithm for hadamard transform. Mathematical Proceedings of the Cambridge Philosophical Society, 1968. 18